# Parametrizing Convex Sets Using Sublinear Neural Networks

## Abstract

We propose a neural parameterization of convex sets by learning sublinear (positively homogeneous and convex) functions. Our networks implicitly represent both the support and gauge functions of a convex body. We prove a universal approximation theorem for convex sets under this parametrization. Empirically, we demonstrate the method on shape optimization and inverse design tasks, achieving accurate reconstruction of target shapes.

## 1 Introduction

Convex sets play a fundamental role in mathematics, optimization, and scientific computing. They arise naturally in applications such as engineering design, inverse problems or computer vision. In theoretical shape optimization, convexity is often assumed to ensure existence of an optimizer. Consequently, the ability to represent, learn, and optimize convex sets efficiently is of broad interest across several disciplines, especially in theoretical shape optimization where it can be used to help generate conjectures on the optimality of certain shapes.

Designing flexible neural representations of convex sets, however, remains challenging. While modern neural networks can model highly complex geometries, they typically do not preserve convexity by construction. Conversely, classical parametrizations based on support or gauge functions can in some cases guarantee convexity but often rely on constrained optimization procedures and are not naturally integrated into modern automatic differentiation frameworks. This raises the following question: can one design a neural architecture whose outputs are convex sets by construction, while remaining expressive enough to approximate arbitrary convex bodies and enabling efficient computation of geometric and PDE-dependent quantities? In this work, we answer this question positively by introducing a neural parametrization based on sublinear networks, which exactly encodes convexity while remaining fully differentiable and amenable to gradient-based optimization.

We first review existing approaches from convex shape optimization and neural geometric representations, highlighting the limitations that motivate our construction.

### 1.1 Convex Shape Optimization

Shape optimization under convexity constraints has been extensively studied due to its analytical tractability and practical relevance (Buttazzo and Guasoni, 1997). Early numerical approaches enforced convexity through penalization, notably via the distance to the convex hull (Oudet, 2004). Alternative formulations characterized convex sets as intersections of half-spaces (Lachand-Robert and Oudet, 2006), which guarantee convexity but limit the representation of smooth geometries.

A widely adopted paradigm relies on functional parametrization using support or gauge functions (Oudet, 2013). These approaches enable compact representations and have been successfully applied in various contexts (Antunes and Bogosel, 2022; Bayen and Henrion, 2012; Bogosel, 2023; Bogosel et al., 2024; Ftouhi, 2025). However, convexity is encoded through second-order differential inequalities, leading to constrained optimization problems that are difficult to handle, especially in higher dimensions. While exact enforcement is possible in two dimensions (Bayen and Henrion, 2012; Bogosel, 2023), extensions to three dimensions always rely on relaxations that may compromise convexity (Antunes and Bogosel, 2022) (Lamberg and Kaasalainen, 2001).

Geometric approaches provide an alternative by directly constraining admissible deformations. For instance, (Bartels and Wachsmuth, 2020) proposes a triangulation-based framework ensuring convexity preservation through restrictions on the deformation field, while (Chakib et al., 2024) makes use of Minkowski deformations.

## 1.2 Neural Representations of Convex Sets

In the last years, a significant amount of work has been devoted to interfacing or replacing classical topology optimization methods with neural networks, parametrizing the shape using a neural network or making use of Physics Informed Neural Networks (PINNs) to solve the underlying state equation (see e.g. (Shin et al., 2023) for a review). Recent advances in machine learning have introduced flexible neural representations of shapes with built-in properties. For instance, (Bélières Frendo et al., 2025) enforces a volume constraint using SympNets; convexity can be enforced at the level of level-set functions (Martinet and Bungert, 2025) using for instance input-convex neural networks (ICNNs) (Amos et al., 2017).

Beyond shape optimization, convex shape priors have been used as an inductive bias in image segmentation (Liu et al., 2025) or for convex decomposition through diffuse interface formulations such as phase-field models (Deng et al., 2020). In (Tětková et al., 2025), it is shown that convexity is an important property of the latent representation of decision regions.

## 1.3 Main contributions

Despite these advances, classical approaches face significant challenges, like exact enforcement of constraints (especially in three dimensions) or flexibility when it comes to the computation of shape-related quantities. On the other hand, neural approaches often rely on implicit parametrization that makes the computation of boundary quantities expensive, since one needs to reconstruct the boundary at every optimization step.

In contrast, our approach aims to combine the strengths of these paradigms by providing a representation based on the gauge and support functions of convex sets that:

1. ensures exact convexity without constraints,

2. can represent any convex set,

3. allows for simple and precise computation of higher-order geometric terms,

4. is able to seamlessly treat different dimensions.

We demonstrate these claims on various inverse and shape optimization problems, like shape reconstruction, the maximization of the torsional rigidity of a rod and classical problems from convex geometry such as the Minkowski and Mahler problems..

From a machine learning perspective, our approach can be interpreted as defining a hypothesis class of neural networks that encode convexity exactly by construction. In contrast to ICNNs, which enforce convexity of scalar functions with respect to inputs, our architecture directly parametrizes convex bodies. This provides an inductive bias tailored to geometric learning problems, enabling both expressive representations and efficient computation of shape-dependent quantities.

## 1.4 Which representation should be used?

In this work, the gauge and support functions representations are most often interchangeable, since the computation of shape-related quantites are made agnostic with respect to the choice of representation (see Section 3). However, there is certain cases where one of the parametrization is preferable, like the gauge parametrization in Section 4.1 or the support parametrization in Section 4.4. The gauge parametrization may also be preferred when one needs an explicit inverse, as illustrated in Section F. Moreover, certain geometric quantities that we do not discuss here are easier to compute with a certain representation (for instance, the *width* is easily computed in the support function parametrization).

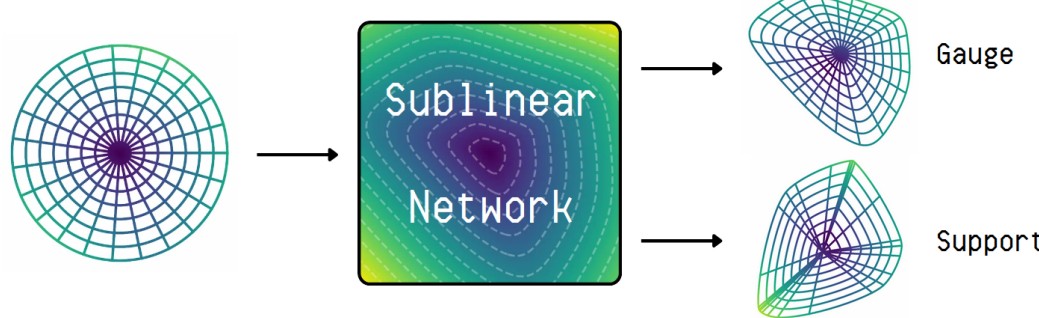

Figure 1: A sublinear network can represent both the gauge and support functions of convex sets, and allow to parameterize smooth bijections that sends the ball on convex sets.

## 2 Parametrization of convex sets using neural networks

### 2.1 Gauge and support functions

In what follows, $\mathcal{K}$ represents the set of convex bodies (i.e. compact convex sets with non-empty interior) in $\mathbb{R}^d$ (endowed with the usual scalar product and Euclidean norm $\|.\|$) which contains 0 in their interior. The gauge and support functions associated to a convex set $\Omega \in \mathcal{K}$ are respectively defined as

$$g_\Omega(u) := \inf \{\lambda \geq 0 : u \in \lambda\Omega\} \qquad \text{and} \qquad h_\Omega(u) := \sup_{x \in \Omega} x \cdot u$$

for $u \in \mathbb{R}^d$. It can readily be seen that both of these functions are *sublinear*. A function $f : \mathbb{R}^d \to R$ is said to be sublinear if it satisfies the two following properties:

**Subadditivity:** for all $x, y \in \mathbb{R}^d$, $f(x + y) \leq f(x) + f(y)$

**Positive homogeneity:** for all $\lambda > 0$ and $x \in \mathbb{R}^d$, $f(\lambda x) = \lambda f(x)$.

Reciprocally, if $f : \mathbb{R}^d \to \mathbb{R}$ is a sublinear function, then it is respectively the gauge and support function of the convex sets (see Theorem A.1)

$$\left\{x \in \mathbb{R}^d : f(x) \leq 1\right\} \qquad \text{and} \qquad \left\{x \in \mathbb{R}^d : x \cdot y \leq f(y) \text{ for all } y \in \mathbb{R}^d\right\} \tag{1}$$

We can use the gauge function of a convex $\Omega$ to give an explicit parametrization of $\Omega$ as the image of the unit ball $B \subset \mathbb{R}^d$ in the following way:

**Proposition 2.1.** *Let $g$ be a positive and sublinear function. The function $\phi : x \longmapsto \frac{\|x\|}{g(x)}x$ (extended by 0 at the origin) is a homeomorphism from $B$ to $\phi(B)$ and the image set $\phi(B)$ is convex with gauge function $g$.*

*Proof.* Let $\Omega := \left\{y \in \mathbb{R}^d : g(y) \leq 1\right\}$. $\Omega$ is convex as the sub-level set of a convex function. For $x \in B$, we have $g(\phi(x)) = \frac{\|x\|}{g(x)}g(x) = \|x\| \leq 1$. Hence, $\phi(B) \subset \Omega$. On the other hand, if $y \in \Omega$, then $x := \phi^{-1}(y) = \frac{g(y)}{\|y\|}y \in B$ and $y = \phi(x) \in \phi(B)$. Therefore, $\phi(B) = \Omega$ is convex and $\phi$ is a homeomorphism. $\square$

Using Proposition A.4 we know that if $\Omega \in \mathcal{K}$ is such that $h_\Omega$ is differentiable on $\mathbb{R}^d \setminus \{0\}$, then $\nabla h_\Omega : \mathbb{S}^{n-1} \to \partial\Omega$ (where $\partial\Omega$ is the boundary of $\Omega$) is a homeomorphism. This leads to the following proposition:

**Proposition 2.2.** *Let $h$ be a positive and sublinear function, differentiable on $\mathbb{R}^d \setminus \{0\}$. The function $\phi : x \longmapsto \|x\|\nabla h(x)$ (extended by 0 at the origin) is bijective from $B$ to $\phi(B)$ and the image set $\phi(B)$ is convex with support function $h$.*

*Proof.* According to the assumptions, there exists $\Omega \in \mathcal{K}$ such that $h = h_\Omega$. We will show that $\Omega = \phi(B)$. Let $y \in \phi(B)$. Then there exists $x \in B$ such that $y = \phi(x)$. If $x = 0$ then $y = 0$ by definition hence $y \in \Omega$ since $h_\Omega$ is positive. If $x \neq 0$ then $y = \|x\|\nabla h_\Omega(x) \in \text{Conv}(\partial\Omega \cup \{0\})$; indeed, since $0 = \phi(0) \in \phi(B)$ and $\|x\| \leq 1$, $y$ is a convex combination of $0$ and $\nabla h_\Omega(x) \in \partial\Omega$ (here, we used that $\nabla h_\Omega$ is 0-homogeneous). Since $\Omega$ is a compact convex set, $\text{Conv}(\partial\Omega \cup \{0\}) = \Omega$ and hence $\phi(B) \subset \Omega$. On the other hand, let $y \in \Omega$. Since $0$ is in the interior for $\Omega$, there exists $\lambda \geq 1$ such that $\tilde{y} := \lambda y \in \partial\Omega$. Hence, there exists $\tilde{x} \in \partial B$ s.t. $\tilde{y} = \nabla h_\Omega(\tilde{x})$. By taking $x = \frac{1}{\lambda}\tilde{x} \in B$, we have $y = \|x\|\nabla h(x)$ and $\Omega \subset \phi(B)$. Hence $\phi(B) = \Omega$, meaning that $\phi(B)$ is convex.

One must now prove the injectivity of $\phi$. Let $x_1, x_2 \in \mathbb{R}^d \setminus \{0\}$ be such that $\phi(x_1) = \phi(x_2)$. Hence, $\nabla h(x_1)$ and $\nabla h(x_2)$ are positively colinear. Moreover, $\nabla h(x_1), \nabla h(x_2) \in \partial\Omega$. However, since $h > 0$, we have that $0 \in int(\Omega)$. Hence, any half line originating at $0$ must intersect $\partial\Omega$ exactly once (by convexity), implying that $\nabla h(x_1) = \nabla h(x_2)$ and further $\|x_1\| = \|x_2\|$. Using the 0-homogeneity of $\nabla h$, we have $\nabla h\left(\frac{x_1}{\|x_1\|}\right) = \nabla h\left(\frac{x_2}{\|x_2\|}\right)$ and using that $\nabla h : \mathbb{S}^{n-1} \to \partial\Omega$ is a homeomorphism, this means that $\frac{x_1}{\|x_1\|} = \frac{x_2}{\|x_2\|}$: therefore $x_1 = x_2$ and $\phi$ is injective. $\qquad\square$

## 2.2 Sublinear neural networks

In what follows, we parametrize convex sets *via* their gauge or support function. To this end, we will construct a neural network $p_\theta : \mathbb{R}^d \to \mathbb{R}$, with parameters $\theta$, that is sublinear by design.

A natural starting point is the MaxOut layer (Goodfellow et al., 2013), defined as $p_\theta(x) = \max_{1 \leq i \leq N}(w_i \cdot x)$, where $w_i \in \mathbb{R}^d$. This choice is motivated by the classical characterization of sublinear functions as point-wise supremum of linear forms, and has already been used to parameterizing sublinear functions in (Haddad and Halder, 2023), which also extended this construction to an ICNN-like architecture with ReLU activation function. However, this representation is non-smooth, which can be limiting in applications requiring differential quantities such as normals or curvature, or even for the application of Proposition 2.2. To address this, we could consider to replace the maximum by a smooth approximation, like the LogSumExp (LSE). Unfortunately, the obtained function would not be sublinear anymore. Nevertheless, we are able to recover sublinearity while preserving the trace on the sphere as we will see hereafter. This requires classical tools from convex analysis, mostly taken from (Schneider, 2013), that we provide in Section A for completeness. The results can also be found in the classical reference (Rockafellar, 1997). We first recall the definition of the convex conjugate of a function:

**Definition 2.1** (Convex conjugate)**.** *Let $f : \mathbb{R}^d \to (-\infty, +\infty]$. The convex conjugate of $f^* : \mathbb{R}^d \to (-\infty, +\infty]$ is defined as*

$$f^*(y) := \sup_{x \in dom(f)} \{y \cdot x - f(x)\} \tag{2}$$

*where $dom(f)$ is the set of $x \in \mathbb{R}^d$ such that $f(x) < +\infty$.*

**Proposition 2.3.** *Let $f : \mathbb{R}^d \to (-\infty, +\infty]$ be a proper convex and closed function such that $f^* \leq 0$ on its domain. Let $x \in \mathbb{R}^d$ and define*

$$h(x) := \|x\|f\left(\frac{x}{\|x\|}\right)$$

*with $h(0) := 0$. Then $h$ is sublinear.*

*Proof.* First, $h$ is positively homogeneous by construction. Hence, sublinearity is equivalent to convexity, and we will show the latter. According Proposition A.1, we can write

$$f(x) = f^{**}(x) = \sup_{y \in \text{dom}(f^*)} \{y \cdot x - f^*(y)\},$$

which leads to $h(x) = \sup_{y \in \text{dom}(f^*)} \{y \cdot x - \|x\|f^*(y)\}$. Since $f^* \leq 0$ on its domain, the function $x \mapsto y \cdot x - \|x\|f(y)$ is convex, implying that $h$ is convex as the supremum of a family of convex functions. $\qquad\square$

Using the previous proposition, we can define our neural network as

$$p_\theta(x) := \beta\|x\|\text{LSE}\left(W^T \frac{x}{\|x\|}\right) \qquad \text{where} \qquad \text{LSE}(y_1, \ldots, y_n) = \log\left(\sum_{i=1}^{n} e^{y_i}\right) \tag{3}$$

and $\theta = (\beta, W) \in \mathbb{R} \times \mathbb{R}^{d \times n}$. In what follows, we assume that $n > d$ and $W$ has full rank $d$.

**Proposition 2.4.** *$p_\theta$ as defined in Eq. (3) is a sublinear function which is $C^\infty$ on $\mathbb{R}^d \setminus \{0\}$.*

*Proof.* The regularity assumption is obvious. Since the LSE is a convex function with values in $\mathbb{R}$, it is proper and closed. The sublinearity can be shown as follows: let $f(x) = \mathrm{LSE}(W^T x)$. According to Proposition 2.3, it is enough to show that $f^*$ is non positive. According to Proposition A.3 and Proposition A.2, we have that

$$f^*(y) \leq W \rhd (-S)(y) = \inf_{x \in \mathbb{R}^n, Wx = y} -S(x) \leq 0$$

where $S$ is the entropy function defined in Proposition A.3. The infimum is taken over the set $Wx = y$ which is non-empty due to the assumption that $W$ is of full rank. $\qquad\square$

**Remark 2.1.** *According to the previous derivations, one does not need to restrict to the LSE activation function. Indeed, any convex function such that $f^* \leq 0$ on its domain would also give rise to a sublinear network. The proof of the universal approximation property (given hereafter) however heavily depends on the fact that the LSE approximates the maximum function.*

**Remark 2.2.** *One might ask whether this construction can be extended to deeper networks. However, such a generalization is not straightforward, due to the complex interaction between convex conjugation and function composition.*

Note that the "extension by homogeneity" of Proposition 2.3 has also been considered in the recent paper (Olausson et al., 2026). However, in their case, the function $f$ is given by an ICNN which does not necessarily guarantees that $f^* \leq 0$ and hence can break the convexity of the extension $h$, as shown by taking $f(x) \equiv -1$.

## 2.3 Universal approximation

According to Proposition 2.1 and Proposition 2.2, given the the previously defined neural network $p_\theta$, we can define the auxiliary neural networks

$$\phi_\theta^g(x) := \frac{\|x\|}{p_\theta(x)} x \tag{4}$$

and

$$\phi_\theta^s(x) := \|x\| \nabla p_\theta(x) \tag{5}$$

which define the convex sets $\Omega_\theta^g := \phi_\theta^g(B)$ and $\Omega_\theta^s := \phi_\theta^s(B)$ (see Fig. 1 for an illustration of the action of these maps on the unit ball). In the case of parametrization equation 4 (resp. equation 5), $p_\theta$ is the gauge function of $\Omega_\theta^g$ (resp. the support function of $\Omega_\theta^s$. We will simply write $\Omega_\theta = \phi_\theta(B)$ when the precise representation (gauge or support) does not matter. In what follows,

$$\mathcal{K}^g := \left\{ \Omega_\theta^g \subset \mathbb{R}^d : \beta > 0, W \in \mathbb{R}^{d \times m}, m > d, \mathrm{rank}(M) = d \right\}$$

will denote the set of convex bodies that are represented by the gauge parameterization Eq. (4) and

$$\mathcal{K}^s := \left\{ \Omega_\theta^s \subset \mathbb{R}^d : \beta > 0, W \in \mathbb{R}^{d \times m}, m > d, \mathrm{rank}(M) = d \right\}$$

the set of convex bodies represented by the support parameterization Eq. (5).

It is of core interest to know whether every convex set can be approximated using this architecture. To this purpose, we will use the Hausdorff distance between two convex sets $K, L \in \mathcal{K}$ which is defined as

$$d_H(K, L) := \sup \left\{ \sup_{x \in K} d(x, L), \sup_{x \in L} d(x, K) \right\}$$

It turns out that we have the following quantitative universal approximation properties in the Hausdorff sense:

**Theorem 2.1.** $\mathcal{K}^s$ *is dense in* $\mathcal{K}$ *with respect to the Hausdorff distance. More precisely, there exists a dimensional constant* $c_d$ *such that for all* $\Omega \in \mathcal{K}$, *all* $\varepsilon > 0$ *small enough and all* $m \geq c_d \left( \frac{diam(\Omega)}{\varepsilon} \right)^{\frac{d-1}{2}}$, *there exists* $\theta = (\beta, W) \in \mathbb{R} \times \mathbb{R}^{d \times m}$ *such that* $d_H(\Omega_\theta^s, \Omega) < \varepsilon$.

**Theorem 2.2.** $\mathcal{K}^g$ *is dense in* $\mathcal{K}$ *with respect to the Hausdorff distance. More precisely, there exists a dimensional constant* $c_d$ *such that for all* $\Omega \in \mathcal{K}$, *all* $\varepsilon > 0$ *small enough and all* $m \geq c_d \left( \frac{diam(\Omega)}{\varepsilon} \right)^{\frac{d-1}{2}}$, *there exists* $\theta = (\beta, W) \in \mathbb{R} \times \mathbb{R}^{d \times m}$ *such that* $d_H(\Omega_\theta^g, \Omega) < \varepsilon$.

In particular, those theorems implies that every convex set can be approximated at the precision of $\varepsilon$ by a network of a size of $O\left(\varepsilon^{-\frac{d-1}{2}}\right)$ neurons. The proof of Theorem 2.1 is based on the following theorem by Bronshteyn and Ivanov (1975):

**Theorem 2.3.** *There exists a constant* $C_d$ *such that for all* $\Omega \in \mathcal{K}$, *there exists* $w_1, \ldots, w_m \in \mathbb{R}^d$ *such that*

$$d_H(\Omega, P_m) \leq C_d \frac{diam(\Omega)}{m^{\frac{2}{m-1}}}$$

*where* $P_N = Conv(w_1, \ldots, w_m)$.

Moreover, we will use the classical fact that for $z_1, \ldots, z_m \in \mathbb{R}$ and $\beta > 0$, we have

$$\max_i z_i \leq \beta \mathrm{LSE}\left( \frac{z_1}{\beta}, \ldots, \frac{z_m}{\beta} \right) \leq \max_i z_i + \beta \log(m). \tag{6}$$

*Proof of Theorem 2.1.* Let $\Omega \in \mathcal{K}$, $\varepsilon > 0$. According to Theorem 2.3, there exist a polytope $P_m = \mathrm{Conv}(w_1, \ldots, w_m)$ such that $d_H(\Omega, P_m) \leq C_d \frac{\mathrm{diam}(\Omega)}{m^{\frac{2}{m-1}}}$. By chosing $m \geq \left( \frac{2 C_d \mathrm{diam}(\Omega)}{\varepsilon} \right)^{\frac{d-1}{2}}$, we get that $d_H(\Omega, P_m) \leq \varepsilon/2$.

It is straightforward to show that the support function of $P_m$ is given by $h_{P_m}(x) := \sup_{1 \leq i \leq m} w_i \cdot x$. Now, let $p_\theta$ be as defined in Eq. (3) with $\theta = (\beta, W_\beta)$ where $W_\beta := \beta^{-1}(w_1 \ldots w_m)^T$. Let $\Omega_\theta^s$ be the associated convex set, defined by Eq. (5). This implies in particular that $h_{\Omega_\theta^s} = p_\theta$.

Now, using the triangle inequality, $d_H(\Omega, \Omega_\theta) \leq d_H(\Omega, P_m) + d_H(P_m, \Omega_\theta)$. According to (Schneider, 2013, Lemma 1.8.14) $d_H(P, \Omega_\theta^s) = \|h_P - p_\theta\|_{C(\mathbb{S}^{n-1})}$. Using Eq. (6) we deduce that

$$\|h_{P_m} - p_\theta\|_{C(\mathbb{S}^{n-1})} \leq \beta \log(m).$$

By taking $\beta \leq \frac{\varepsilon}{2 \log(m)}$, we have that

$$d_H(K, \Omega_\theta^s) \leq d_H(K, P) + d_H(P, \Omega_\theta^s) \leq \varepsilon$$

and the result follows. $\qquad \square$

Now let us turn to the proof of Theorem 2.2. While Theorem 2.3 controls the number of vertices of a polytope approximating a convex set, we will need an analogous result to control the number of faces. This is the object of a theorem of Dudley (1974):

**Theorem 2.4.** *There exists a constant* $C_d$ *such that for all* $\Omega \in \mathcal{K}$, *there exists* $w_1, \ldots, w_m \in \mathbb{R}^d$ *such that*

$$d_H(\Omega, P_m) \leq C_d \frac{diam(\Omega)}{m^{\frac{2}{m-1}}}$$

*where*

$$P_m = \bigcap_{1 \leq i \leq m} \left\{ x \in \mathbb{R}^d : w_i \cdot x \leq 1 \right\}.$$

We can now show the proof of Theorem 2.2, which uses the same ideas than Theorem 2.1 but involves a few additional technicalities.

*Proof of Theorem 2.2.* We first prove the following fact: for $K, L \in \mathcal{K}$, we have $d_H(K, L) \leq \|\rho_K - \rho_L\|_{C(\mathbb{S}^{n-1})}$ where

$$
\rho_K \colon \mathbb{S}^{n-1} \longrightarrow \mathbb{R}^+
$$
$$
x \longmapsto \sup\{r > 0 : rx \in K\}
$$

is the *radial function* of $K$. In particular, it is immediate that the radial function is the inverse of the gauge function on $\mathbb{S}^{n-1}$. For $x \in K$, let us define

$$
y = \begin{cases} \rho_L \left( \frac{x}{\|x\|} \right) \frac{x}{\|x\|} & \text{if } x \notin L \\ x & \text{if } x \in L \end{cases}.
$$

Then $y \in L$ and we have

$$
d(x, L) \leq \max \left\{ 0, \|x\| - \rho_L \left( \frac{x}{\|x\|} \right) \right\} \leq \left| \rho_K \left( \frac{x}{\|x\|} \right) - \rho_L \left( \frac{x}{\|x\|} \right) \right| \leq \|\rho_K - \rho_L\|_{C(\mathbb{S}^{n-1})}
$$

Hence $\sup_{x \in K} d(x, L) \leq \|\rho_K - \rho_L\|_{C(\mathbb{S}^{n-1})}$ . By symmetry, we deduce the claim.

Now, let $\Omega \in \mathcal{K}$, $\varepsilon > 0$. According to Theorem 2.4, there exist a polytope $P_m = \bigcap_{1 \leq i \leq m} \left\{ x \in \mathbb{R}^d : w_i \cdot x \leq 1 \right\}$ such that $d_H(\Omega, P_m) \leq C_d \frac{\text{diam}(\Omega)}{m^{\frac{2}{m-1}}}$. By chosing $m \geq \left( \frac{2C_d \text{diam}(\Omega)}{\varepsilon} \right)^{\frac{d-1}{2}}$, we get that $d_H(\Omega, P_m) \leq \varepsilon/2$.

In particular, we have $g_{P_m}(x) = \max_{1 \leq i \leq m} w_i \cdot x$. For $\varepsilon$ small enough, $d_H(\Omega, P_m) < \varepsilon/2$ implies that $\min_{u \in \mathbb{S}^{n-1}} g_{P_m}(u) \geq \frac{\alpha_\Omega}{2}$ where $\alpha_\Omega := \min_{u \in \mathbb{S}^{n-1}} g_\Omega(u) > 0$ . Similarly as before, for any $\delta > 0$, we can take $\beta \leq \frac{\alpha_\Omega^2}{16 \log(m)} \varepsilon$ small enough so that $\min_{u \in \mathbb{S}^{n-1}} p_\theta(u) \geq \frac{\alpha_\Omega}{4}$. Hence, for $\Omega_\theta^g$ defined as in Eq. (4) with $\theta = (\beta, W_\beta)$, $W_\beta = \left( \frac{w_1}{\beta} \ldots \frac{w_m}{\beta} \right)^T$, we have

$$
\begin{aligned}
\|\rho_{\Omega_\theta^g} - \rho_{P_m}\|_{C(\mathbb{S}^{n-1})} &= \left\| \frac{1}{p_\theta} - \frac{1}{g_{P_m}} \right\|_{C(\mathbb{S}^{n-1})} \\
&\leq \frac{\|g_{P_m} - p_\theta\|_{C(\mathbb{S}^{n-1})}}{\min_{\mathbb{S}^{n-1}} p_\theta \min_{\mathbb{S}^{n-1}} g_{P_m}} \\
&\leq \frac{\beta \log(m)}{\min_{\mathbb{S}^{n-1}} p_\theta \min_{\mathbb{S}^{n-1}} g_{P_m}} \qquad \text{using Eq. (6)} \\
&\leq \frac{8\beta \log(m)}{\alpha_\Omega^2} \leq \frac{\varepsilon}{2}
\end{aligned}
$$

leading to $d_H(\Omega, \Omega_\theta^g) \leq d_H(\Omega, P_m) + d_H(P_m, \Omega_\theta^g) \leq d_H(\Omega, P_m) + \|\rho_{\Omega_\theta^g} - \rho_{P_m}\|_{C(\mathbb{S}^{n-1})} \leq \varepsilon.$ $\qquad \square$

**Symmetries** A natural question is to know whether we can enforce symmetries on the parametrized shapes. In term of group actions, it amounts at asking if, for a certain group of symmetries $G$ we can make $\Omega_\theta$ *invariant* with respect to the action of $G$, i.e. $g.\Omega_\theta = \Omega_\theta$ for all $g \in G$. In our case, we achieve invariance by *frame averaging* (Puny et al., 2021). Given $G$ a finite subgroup of isometries of $\mathbb{R}^d$ and $p_\theta$ a sublinear network, we define $p_\theta^G(x) := \frac{1}{|G|} \sum_{\tilde{g} \in G} p_\theta(\tilde{g}.x)$. we have the following proposition:

**Proposition 2.5.** *Let $p_\theta^G$ be either the gauge or support function of a convex set $\Omega_\theta$. Then $\Omega_\theta$ is invariant with respect to $G$.*

*Proof.* We will use Eq. (1). Let $p_\theta$ be a sublinear network.

**Case 1 (gauge):** Let $\Omega_\theta := \left\{ x \in \mathbb{R}^d : p_\theta^G(x) \leq 1 \right\}$. For $x \in \Omega_\theta$ and $g \in G$, we have that $p_\theta^G(g.x) = \frac{1}{|G|} \sum_{\tilde{g} \in G} p_\theta(g.(\tilde{g}.x)) = \frac{1}{|G|} \sum_{\tilde{g} \in G} p_\theta((g\tilde{g}).x) = \frac{1}{|G|} \sum_{\tilde{g} \in G} p_\theta(\tilde{g}.x) = p_\theta^G(x) \leq 1$ (where we used the change of variable $\tilde{g} \leftarrow \tilde{g}g$) and hence $g.\Omega_\theta \subset \Omega_\theta$. We can then deduce the reverse inclusion by $x \in \Omega_\theta \implies g^{-1}.x \in \Omega_\theta \implies x \in g.\Omega_\theta$.

**Case 2 (support):** Let $\Omega_\theta := \left\{ x \in \mathbb{R}^d : x \cdot y \leq p_\theta^G(y) \text{ for all } y \in \mathbb{R}^d \right\}$. For $x \in \Omega_\theta$ and $g \in G$, we have for all $y \in \mathbb{R}^d$:

$$(g.x) \cdot y = x \cdot (g^{-1}.y) \leq p_\theta^G(g^{-1}.y) = p_\theta^G(y)$$

hence $g.\Omega_\theta \subset \Omega_\theta$. The reverse inclusion follows. $\qquad\square$

An experiment making use of the symmetries of both the gauge and support functions can be found in Section 4.5.

## 3 Computation of shape quantities

Shape optimization problems, as well as certain inverse problems, require to optimize a certain criterium involving quantities related to the shape, like the volume, perimeter, curvature, etc, for which we can benefit from automatic differentiation. Other geometric quantities have also been explored in the literature (Martinet and Ftouhi, 2026).

Let $p_\theta$ be a sublinear neural network as defined in Eq. (3). In what follows, $\phi_\theta$ will denote either the neural network defined in Eq. (4) or in Eq. (5) *via* $p_\theta$. We will reformulate shape optimization problems into finite-dimensional optimization problems over the parameters $\theta$ of $\Omega_\theta = \phi_\theta(B)$. Using the previous propositions, we have the guarantee that $\Omega_\theta$ will stay convex in the course of the optimization. Moreover, since $\phi_\theta$ is a smooth homeomorphism, we will be able to compute integral quantities (like the volume or the surface area) by using the change of variable formula.

### 3.1 Integral quantities

We pull back the computations to the reference domain $B$, using change of variables (Evans and Gariepy, 2025):

$$\int_{\Omega_\theta} f dx = \int_B (f \circ \phi_\theta) \mathrm{Jac}(\phi_\theta) dx \quad \text{and} \quad \int_{\partial\Omega_\theta} g d\sigma = \int_{\partial B} (g \circ \phi_\theta) \mathrm{Jac}_{\partial B}(\phi_\theta) d\sigma,$$

where $\mathrm{Jac}(\phi_\theta) = |\det(D\phi_\theta)|$, $\mathrm{Jac}_{\partial B}(\phi_\theta) = \mathrm{Jac}(\phi_\theta) \| (D\phi_\theta)^{-T} n_B \|$ and $n_B(x) := x$ on $\partial B$ is the unit outward normal vector. For instance, we can compute the volume and perimeter (i.e. surface area) of $\Omega_\theta$ as

$$\mathrm{Vol}(\Omega_\theta) = \int_B \mathrm{Jac}\phi_\theta dx \quad \text{and} \quad \mathrm{Per}(\Omega_\theta) = \int_{\partial B} \mathrm{Jac}_{\partial B}(\phi_\theta) d\sigma$$

All the differential quantities are automatically computed using PyTorch (Paszke et al., 2019). The integrals can be discretized using either Monte-Carlo or fixed quadrature points. We chose the latter approach, as we will mainly use L–BFGS as an optimizer, which is known for being sensitive to noise. In particular, we use a Fibonacci lattice approach for the discretizations on the 2-sphere (González, 2010).

### 3.2 Geometric-differential quantities

**Normal vector:** Let $y = \phi_\theta(x)$, $x \in \partial B$. We can express the normal vector at $y \in \partial\Omega_\theta$ by

$$n_\theta(y) = \frac{(D\phi_\theta)^{-T}(x)n_B(x)}{\left\| (D\phi_\theta)^{-T}(x)n_B(x) \right\|}.$$

Indeed, if $\varphi$ is a level set function associated to a smooth set $\Omega$ (i.e. $\Omega = \{\varphi \leq 0\}$), then for $y \in \partial\Omega$, the normal vector can be computed as

$$n(y) = \frac{\nabla\varphi(y)}{\|\nabla\varphi(y)\|}.$$

In the case of $\Omega_\theta$, the function $\varphi(y) = \|\phi_\theta^{-1}(y)\|^2 - 1$ is an admissible level set function since $\varphi(y) \leq 0 \iff y \in \Omega_\theta$. Computing the gradient leads to $\nabla\varphi(y) = D\left(\phi_\theta^{-1}\right)^T(y).2\phi_\theta^{-1}(y) = 2\left(D\phi_\theta\right)^{-T}(x)x$, from which we deduce the formula by normalization. Notice that we never need the expression of $\phi_\theta^{-1}$ to compute $n_\theta(y)$ when $y = \phi_\theta(x)$.

**Curvature terms:** The *Weingarten map* at $y$ is defined as

$$S_y : T_y \partial \Omega_\theta \longrightarrow T_y \partial \Omega_\theta$$
$$v \longmapsto D_\Gamma n(y).v$$

where $D_\Gamma n(y)$ is the tangential derivative of $n$, i.e. the restriction of $Dn(y) \in \mathbb{R}^{d \times d}$ to the tangent space $T_y \partial \Omega_\theta$. Using this map, we can respectively define the mean curvature and Gaussian curvature as

$$H_\theta(y) = (d-1)^{-1} \operatorname{tr} S_y \qquad \text{and} \qquad \kappa_\theta(y) = \det S_y$$

In order to compute the Weingarten map numerically, one introduces the Housholder matrix $H_y = I - vv^T$ where

$$v = \frac{n(y) - e_d}{\|n(y) - e_d\|},$$

$e_1, \ldots, e_d$ being the canonical basis of $\mathbb{R}^d$. Hence, $H_y n = e_d$ and in particular, the first $d-1$ lines of $H_y$ are orthogonal to $n(y)$ and hence forms an orthonormal basis of $T_y \partial \Omega_\theta$. One can then compute $D_\Gamma n(y)$ in this basis as

$$D_\Gamma n(y) = \tilde{H}_y Dn(y) \tilde{H}_y^T \in \mathbb{R}^{(d-1) \times (d-1)}$$

where $\tilde{H} = (H_{ij})_{\substack{1 \le i \le d-1 \\ 1 \le j \le d}}$.

### 3.3 PDE-related quantities

It is common in shape optimization to consider quantities that depends on the solution of a PDE. For instance, optimal design of structures often minimize for the compliance, which is a by-product of the linear elasticity equation; aerodynamic shape optimization needs to solve Navier-Stokes. In this section, we show how it is possible to seamlessly bridge precise and robust mesh free methods with the auto-differentiation of PyTorch in order to easily compute derivatives of PDE-dependent quantities in the simple case of the Poisson equation.

**Mesh free Galerkin method:** For $f \in L^2(\Omega_\theta)$, the Poisson problem with Dirichlet boundary condition aims at finding $u \in H_0^1(\Omega_\theta)$ which satisfies

$$\begin{cases} -\Delta u &= f \quad \text{in } \Omega_\theta, \\ u &= 0 \quad \text{on } \partial \Omega_\theta. \end{cases} \tag{7}$$

Passing to the *weak formulation* (Evans, 2022) and changing variables, Eq. (7) can be express in a weak sense as

$$\int_B A_\theta \nabla u \cdot \nabla v = \int_B (\operatorname{Jac}\phi_\theta)(f \circ \phi_\theta)v,$$

for $u, v \in H_0^1(B)$, where $A_\theta := (\operatorname{Jac}\phi_\theta)(D\phi_\theta)^{-1}(D\phi_\theta)^{-T}$. The Dirichlet boundary condition is weakly enforced by adding a penalization term with penalty parameter $\alpha > 0$. The resulting problem is then discretized on a subspace spanned by radial basis functions (RBFs) $\varphi_i(x) := \psi(|x - x_i|)$, $x_i \in B$, possibly augmented by a polynomial basis (Wendland, 1999) leading to a system $K\bar{u} = \bar{f}$, where

$$K_{ij} = \int_B A_\theta \nabla \varphi_i \cdot \nabla \varphi_j + \alpha \int_{\partial B} \varphi_i \varphi_j \qquad \text{and} \qquad \bar{f}_i = \int_B (\operatorname{Jac}\phi_\theta)(f \circ \phi_\theta)\varphi_i.$$

The integrals are discretized as previously described. We can then solve the previous linear system and recover the solution $u(x) = \sum_i \bar{u}_i \varphi(x)$.

**Remark 3.1.** *The RBF-Galerkin method is chosen because of its theoretical convergence guarantees (unlike other meshless methods relying on the strong formulation of the PDE like the Kansa method (Fasshauer, 2007) or PINN-based methods (Dissanayake and Phan-Thien, 1994)) and is easy to implement in PyTorch.*

**Method of fundamental solutions**  An important particular case of Eq. (7) is for $f = 1$, in which case the equation can be solved thanks to a method exhibiting spectral convergence. The details are provided in Section C.

**A note on the use of PINNs**  One may wonder why we do not use PINNs to solve the PDEs, as in related work (Bélières Frendo et al., 2025). Our choice is guided by both practical and structural considerations. On the practical side, recent studies indicate that PINNs do not match the efficiency of classical solvers on several low-dimensional PDEs (Grossmann et al., 2024; McGreivy and Hakim, 2024). On the structural side, while (Bélières Frendo et al., 2025) relies on a min–min formulation enabling joint optimization, most shape optimization problems are naturally min–max, requiring the inner problem to be (approximately) solved at each step, which increases computational cost.

For completeness, we include a comparison with PINNs in Section D, where classical methods remain way more efficient in our setting.

**On shape derivatives and the FEM**  Classical approaches to minimizing a shape functional rely on shape derivatives evaluated on a mesh. Their numerical implementation typically involves additional components such as adjoint states and extension–regularization procedures, which require solving additional PDEs on the domain. In contrast, our approach avoids these steps by leveraging automatic differentiation.

That said, mesh-based methods might remain desirable in a context where high accuracy is needed. However, there is currently no standard PyTorch-based framework for finite element computations. We discuss a possible interface with external FEM libraries in Section F.

**Remark 3.2.** *Notably, the arguments in this section do not rely on the convexity of $\Omega_\theta$ or on any special structure of $\phi_\theta$, except than being a diffeomorphism. This suggests that the framework can be extended beyond the present setting to arbitrary invertible neural networks $\phi_\theta$, which we leave to future work.*

## 4  Application to shape optimization problems

In this section, we present the performance of our method on a range of problems arising from geometry and shape optimization. For fairness with respect to classical methods and easy reproducibility, all experiments are conducted on a single CPU (AMD Ryzen 7 Pro) with 28 GB of RAM. The computational time for all two-dimensional cases is on the order of one minute, while in three dimensions it is typically on the order of ten minutes. The implementation is also CUDA-compatible and can therefore benefit from standard GPU acceleration. Contrary to classical numerical convex shape optimization, every shape optimization problems considered here reduces to an unconstrained optimization problem.

Note that a direct quantitative comparison with classical methods of convex shape optimization is not possible in practice, as no publicly available implementations seems to exist in a form that would allow a consistent and reproducible evaluation.

### 4.1  Learning convex sets with noisy boundary samples

The first problem we consider is to reconstruct a convex shape from noisy observations of its boundary. More precisely, given samples $y_1, \ldots, y_n \in \mathbb{R}^d$, we minimize the loss function

$$L(\theta) = \sum_{i=1}^{n} |p_\theta(y_i) - 1|^2$$

where $p_\theta$ is the gauge function of the convex set $\Omega_\theta$. This loss is motivated by the fact that $\partial\Omega_\theta = \{p_\theta = 1\}$.

In practice, the observations $y_i$ are generated synthetically as follows: first, $x_i \sim \mathcal{U}(\partial B)$. Then, $y_i = \phi_{\text{target}}(x_i) + \varepsilon_i$ where $\varepsilon_i \sim \mathcal{N}(0, \sigma)$, and $\phi_{\text{target}}$ denotes a maps from $B$ to $\Omega_{\text{target}}$. The results, shown in Fig. 2, are obtained with $n = 1000$ samples and varying noise levels across three different shapes. We observe that the convex inductive bias enables an accurate reconstruction of the shapes, even with a relatively small

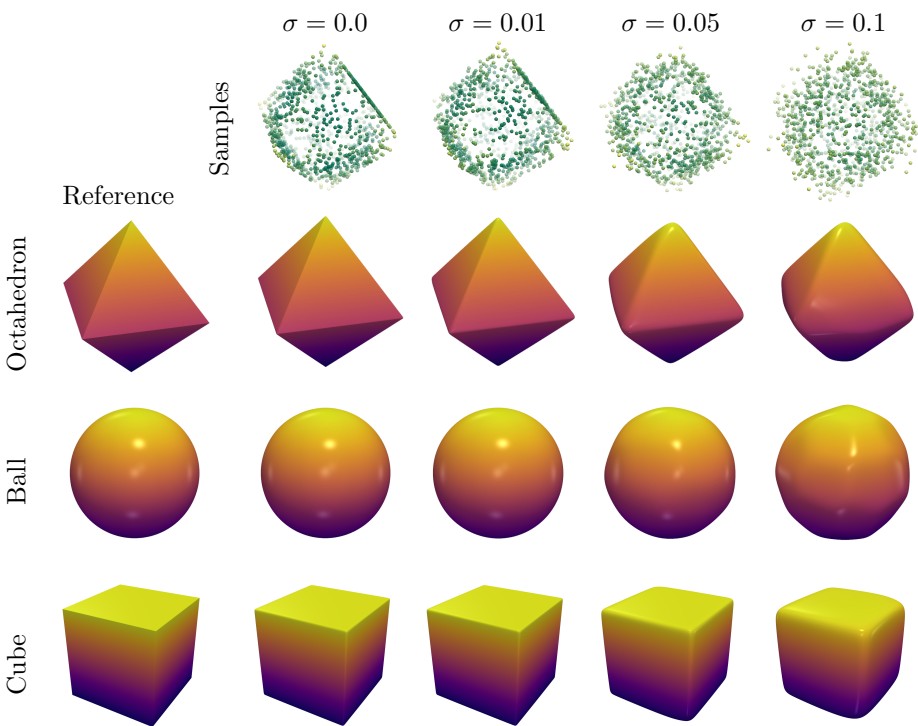

Figure 2: Reconstruction of convex shapes from noisy point clouds. First column: reference shapes. Top row: input point clouds sampled from the octahedron under increasing Gaussian noise levels $\sigma$. Remaining rows: reconstructed shapes for different target geometries.

number of samples and substantial noise. Additional statistical experiments assessing sensitivity to noise and to the amount of samples over multiple runs are reported in Section E.

## 4.2 Optimization of a Poisson problem

One of the simplest PDE-constrained shape optimization problem is the following: considering a function $f : \mathbb{R}^d \to \mathbb{R}$ and $u_\theta$ to be the solution of Eq. (7) on a domain $\Omega_\theta \in \mathcal{K}$, what is the minimum of $J(\Omega_\theta) := \int_{\Omega_\theta} u_\theta$? As described previously, the solution $u_\theta$ is computed by a mesh free Galerkin method, while the integral is evaluated by change of variables. The loss function that we minimize is then

$$L(\theta) = \int_B (u_\theta \circ \phi_\theta) \mathrm{Jac}(\phi_\theta) dx$$

We performs our experiments in the same settings as in (Bogosel, 2023), i.e. in dimension 2 for two different functions $f$. The optimal shapes are given in Fig. 3. They can be compared to the ones in (Bogosel, 2023).

## 4.3 Maximization of the gradient of the torsion function

In linear elasticity, the quantity $\|\nabla u_\Omega\|_{L^\infty}$ (where $u_\Omega$ is the torsion function Eq. (8)) represents the maximal shear stress of a rod with section $\Omega$. In the case of a planar convex domain $\Omega \subset \mathbb{R}^d$, many authors are interested in knowing what shape maximizes this quantity when prescribing the area or the perimeter (see (Burdzy et al., 2025) and references therein). Using homogeneity arguments, it amounts at maximizing respectively $\frac{\|\nabla u_\Omega\|_{L^\infty}}{\mathrm{Vol}(\Omega)^{1/d}}$ and $\frac{\|\nabla u_\Omega\|_{L^\infty}}{\mathrm{Per}(\Omega)^{1/(d-1)}}$. Assuming enough regularity, the maximum principle applied on $|\nabla u|^2$ ensures that the infinity norm is attained on the boundary, i.e. $\|\nabla u_\Omega\|_{L^\infty(\Omega)} = \|\partial_n u\|_{L^\infty(\partial\Omega)}$. For a parametrized convex set $\Omega_\theta$, maximizing the previous quantities is hence equivalent to maximizing respectively

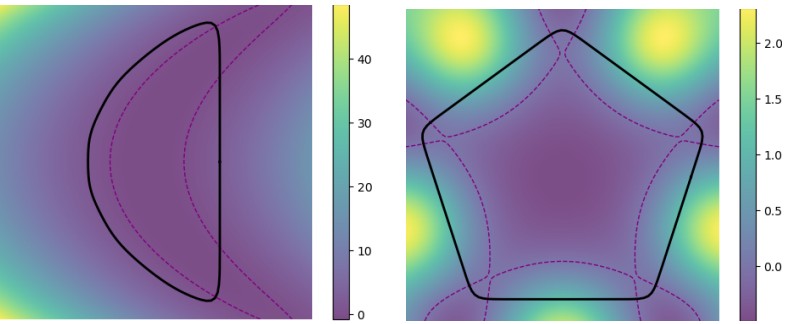

Figure 3: Optimal shape for the Poisson-based shape optimization problem (in black) along with the function $f$. The dotted line is the 0 level set.

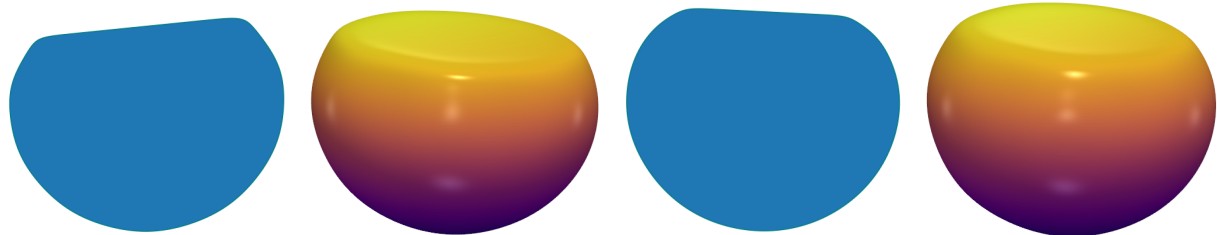

Figure 4: Optimal shapes for maximizing the torsion gradient under two constraints: fixed volume (left) and fixed perimeter (right). While the resulting shapes are similar, the choice of normalization produces distinct geometries.

the two loss functions

$$L_{\text{Vol}}(\theta) := \frac{|\partial_n u_{\Omega_\theta}(\phi_\theta(x))|}{\text{Vol}(\Omega_\theta)^{1/d}} \qquad \text{and} \qquad L_{\text{Per}}(\theta) := \frac{|\partial_n u_{\Omega_\theta}(\phi_\theta(x))|}{\text{Per}(\Omega)^{1/(d-1)}}.$$

for a fixed $x \in \partial B$. The optimal shapes in dimension 2 and 3 are given in Fig. 4. In order to show that our algorithm easily adapts to higher dimensions, we give in Table 1 the optimal values obtained with our method for each functionals in dimension up to 4. In dimension 2, the values that are obtained matches previous numerical experiments found in the literature (Burdzy et al., 2025).

## 4.4 Minkowski problem

The Minkowski problem (Huang et al., 2025) is a foundational problem in convex and differential geometry, connected to the famous Monge–Ampère equation. Informally speaking, it is concerned with the question of existence of a convex set with prescribed Gaussian curvature. More precisely, let $g : \mathbb{S}^{n-1} \to \mathbb{R}$ be a positive, continuous function. Does there exists a convex set $\Omega$ such that $\kappa_\Omega \circ n_\Omega^{-1} = g$ on $\partial B$? It is known that such $\Omega$ exists if and only if $g$ verifies $\int_{\partial B} \frac{u}{g(u)} du = 0$. We can approximate this problem by formulating it as a

Table 1: Optimal values for $L_{\text{Vol}}$ and $L_{\text{Per}}$ in several dimensions.

| Dimension | $L_{\text{Vol}}^*$ | $L_{\text{Per}}^*$ |
|---|---|---|
| 2 | 0.35809 | 0.09886 |
| 3 | 0.30918 | 0.13842 |
| 4 | 0.28397 | 0.15532 |

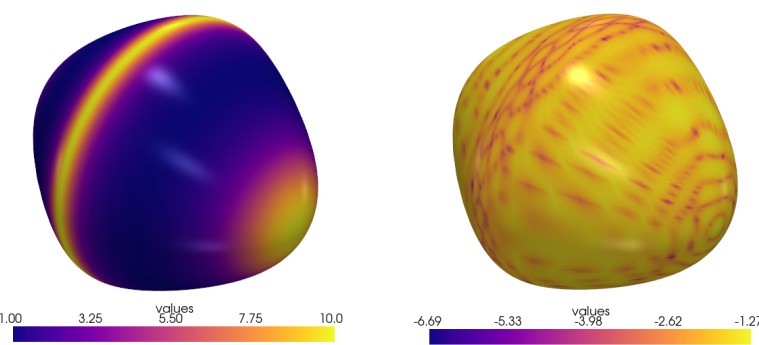

Figure 5: Solution to the Minkowski problem. Left: prescribed curvature. Right: $\log_{10}$ relative error. The method accurately recovers the target curvature distribution.

regression problem, by minimizing the relative mean squared error between the actual and target curvature:

$$L(\theta) = \int_{\partial B} \left| \frac{\kappa_\theta(n_\theta^{-1}(u)) - g(u)}{g(u)} \right|^2 du.$$

Except for $n_\theta^{-1}$, everything is easily computable in the present framework. However, as it is pointed out in (Schneider, 2013), if $p_\theta$ is the support function of $\Omega_\theta$ (i.e., choosing the parametrization Eq. (5)) we have $n_\theta^{-1} = \nabla p_\theta = \phi_\theta$ on $\partial B$. Hence, the previous loss is readily computable. We show that our method successfully applies to the Minkowski problem in Fig. 5. The final $L^2$ relative error is of the order of $10^{-2}$.

### 4.5 Minimization of the Mahler volume

Gauge and support functions are related to each other through the notion of *polar body*. The polar body of a convex set $K \in \mathcal{K}$ is defined as $K^\circ := \left\{ x \in \mathbb{R}^d : x \cdot y \leq 1 \text{ for all } y \in K \right\}$. Then, the gauge function of $K$ is the support function of $K^\circ$ (see (Schneider, 2013, Lemma 1.7.13)), i.e. $g_K = h_{K^\circ}$.

In this section, we are interested in the minimization of the *Mahler volume* which is the product of the volumes of a convex $\Omega$ and its polar body $\Omega^\circ$, i.e. $\mathrm{Vol}_M(\Omega) := \mathrm{Vol}(\Omega)\mathrm{Vol}(\Omega^\circ)$. Our framework allows us to easily compute both volumes using simultaneously the support and the gauge parametrization. The loss function that we minimize is

$$L(\theta) = \left( \int_B \mathrm{Jac}(\phi_\theta^s) dx \right) \left( \int_B \mathrm{Jac}(\phi_\theta^g) dx \right)$$

where $\phi_\theta^s$ and $\phi_\theta^g$ are defined using the same sublinear network $p_\theta$.

In $\mathbb{R}^2$, it is known that the convex set with $n$-fold rotational symmetry that minimizes the Mahler volume is the regular polygon, and the optimal value is $n^2 \sin(\pi/n)^2$ (Böröczky et al., 2013). Using our parametrization of symmetric convex sets, we get the results presented in Fig. 6. As you can see, our parametrization is flexible enough to recover polygons.

**Reproducibility** The code is fully available on GitHub: `https://anonymous.4open.science/r/SublinearNet-5726/`.

## 5 Discussion

The proposed approach allows for a provably expressive and unconstrained representation of convex sets. Profiting from the auto-differentiation capabilities of PyTorch, it allows to easily solve a large variety of shape optimization problems without any shape derivative computation.

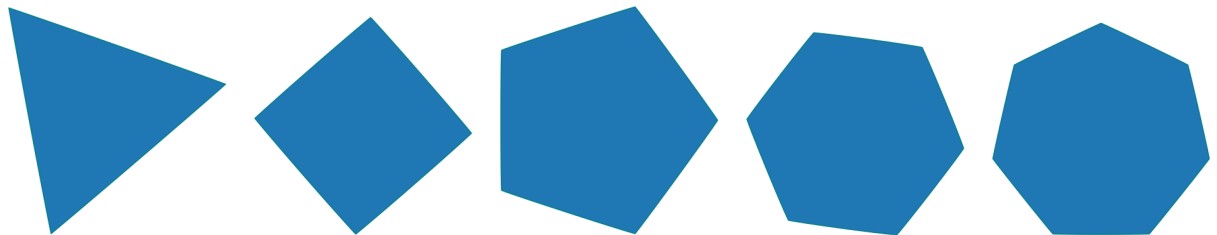

Figure 6: Minimizers of the Mahler volume under $n$-fold symmetry constraints. The method recovers polygonal structures, highlighting its ability to represent non-smooth convex shapes.

**Limitations**   However, several limitations remain. The reliance on quasi-Newton methods like L–BFGS with fixed discretizations limits scalability due to the curse of dimensionality. This could however be mitigated by first training using Monte–Carlo sampling with a first–order optimizer, then fine–tuning using L–BFGS. Certain geometric quantities (e.g., curvature) may become unstable near non-smooth shapes. Memory usage is also higher than in classical approaches, since PDE solvers must remain in the computational graph. Finally, symmetry enforcement increases computational cost linearly with respect to the size of the symmetry group due to repeated evaluations.

**Broader impact**   We validate the proposed method primarily on mathematical shape optimization problems, where it may help accelerate exploration and conjecture generation. Extending the framework to more realistic engineering settings—such as structural or aerodynamic optimization—could further broaden its applicability and impact.

## 6   Conclusion

We introduced a neural parametrization of convex sets based on sublinear networks, ensuring exact convexity while retaining strong approximation capabilities. We proved universal approximation in the Hausdorff distance and showed how the parametrization enables efficient evaluation of geometric and PDE-dependent quantities via automatic differentiation.

Numerical experiments demonstrate competitive performance on a range of shape optimization problems, and show that classical solvers combined with our parametrization outperform PINN-based approaches in both accuracy and efficiency.

Future work includes extending the framework to more complex PDEs (e.g., elasticity or fluid dynamics) and designing architectures that preserve additional geometric constraints, like the volume (using measure-preserving neural networks) or the perimeter, for which a new architecture must probably be derived.

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

## A    Reminder of convex analysis

Here we give some useful results of convex analysis. The following proposition can be found in (Boyd and Vandenberghe, 2004, Exercise 3.39):

**Proposition A.1.** *Let $f : \mathbb{R}^d \to [-\infty, +\infty]$ be a proper convex and closed function. Then $f^{**} = f$.*

The behavior of the convex conjugate with respect to the composition needs the following definition (Bauschke and Combettes, 2020, Definition 12.34):

**Definition A.1** (Infimal postcomposition). *Let $f : \mathbb{R}^d \to [-\infty, +\infty]$ and $L : \mathbb{R}^d \to \mathbb{R}^m$. The infimal postcomposition of $f$ by $L$ is*

$$L \triangleright f : \mathbb{R}^m \longrightarrow [-\infty, +\infty]$$
$$y \longmapsto \inf_{Lx=y} f(x)$$

In our case, we only need the conjugate of the composition with a linear map, which is given by the following proposition (adapted from (Bauschke and Combettes, 2020, Proposition 13.24)):

**Proposition A.2** (Composition with a linear map). *Let $f : \mathbb{R}^d \to (-\infty, +\infty]$ and $A \in \mathbb{R}^{d \times m}$. Then*

$$(f \circ A)^* \leq A^T \triangleright f^*.$$

Finally, we need to know the conjugate of the log-sum-exp function:

**Proposition A.3** (Convex conjugate of the log-sum-exp). *The convex conjugate of the log-sum-exp is the negative entropy function $-S$ where*

$$S : \Delta^N \longrightarrow (0, +\infty)$$
$$y \longmapsto -\sum_i y_i \log y_i$$

*where $\Delta^N$ is the $N$-dimensional simplex.*

This can be shown by computing the optimality conditions in equation 2 (see (Boyd and Vandenberghe, 2004, Example 3.25) for more details).

For the next theorem, see (Schneider, 2013, Theorem 1.7.1):

**Theorem A.1.** *If $f : \mathbb{R}^d \to \mathbb{R}$ is a sublinear function, then there is a unique convex body with support function $f$.*

For the following property, see (Schneider, 2013, Corolary 1.7.3):

**Proposition A.4.** *Let $\Omega$ be a convex body. Its support function $h_\Omega$ is differentiable at $u \in \mathbb{R}^d \setminus \{0\}$ if and only if the support set at $u$ contains only one point $x$. In this case, $x = \nabla h_\Omega(u)$.*

## B  Computation of shape quantities

### B.1  Geometric–differential quantities

**Expression of the normal vector**  As it has been stated, the normal vector is defined by

$$n_\theta\left(y\right) = \frac{(D\phi_\theta)^{-T}(x)n_B(x)}{\left\|(D\phi_\theta)^{-T}(x)n_B(x)\right\|}.$$

for $y = \phi_\theta(x)$, $x \in \partial B$. Indeed, if $\varphi$ is a level set function associated to a smooth set $\Omega$ (i.e. $\Omega = \{\varphi \leq 0\}$), then for $y \in \partial\Omega$, the normal vector can be computed as

$$n(y) = \frac{\nabla\varphi(y)}{\|\nabla\varphi(y)\|}.$$

In the case of $\Omega_\theta$, the function $\varphi(y) = \|\phi_\theta^{-1}(y)\|^2 - 1$ is an admissible level set function since $\varphi(y) \leq 0 \iff y \in \Omega_\theta$. Computing the gradient leads to $\nabla\varphi(y) = D\left(\phi_\theta^{-1}\right)^T(y).2\phi_\theta^{-1}(y) = 2\left(D\phi_\theta\right)^{-T}(x)x$, from which we deduce the formula by normalization. Notice that we never need the expression of $\phi_\theta^{-1}$ to compute $n_\theta(y)$ when $y = \phi_\theta(x)$.

## C  Method of fundamental solutions

An important special case of Eq. (7) is the case where $f$ is identically equal to 1, i.e.

$$\begin{cases} -\Delta u &=& 1 & \text{in } \Omega_\theta, \\ u &=& 0 & \text{on } \partial\Omega_\theta. \end{cases} \tag{8}$$

In this case, the solution $u \in H_0^1(\Omega_\theta)$ is called the *torsion function*, which is an important function in mechanics that has been extensively studied in mathematics. One important derived quantity is the *torsional rigidity* defined as

$$T(\Omega_\theta) = \int_{\Omega_\theta} u \tag{9}$$

which describes the rigidity of a rod of cross section $\Omega_\theta$.

It is well known that Eq. (8) can be solved in an extremely precise and efficient way using the *method of fundamental solutions* (Barnett and Betcke, 2008; Antunes and Bogosel, 2022; Bogosel, 2016). Indeed, by putting $\phi(x) = u(x) + \frac{x_1^2}{2}$, Eq. (8) can be equivalently reformulated as the following Laplace equation:

$$\begin{cases} -\Delta\phi &=& 0 & \text{in } \Omega_\theta, \\ \phi &=& \frac{x_1^2}{2} & \text{on } \partial\Omega_\theta. \end{cases} \tag{10}$$

One may then seek an approximate solution $\tilde{\phi}$ expressed as a linear combination of fundamental solutions, namely,

$$\tilde{\phi}(x) := \sum_{i=1}^{n} c_i\psi(x - y_i),$$

where $\psi$ is the fundamental solution to $-\Delta \psi = \delta_0$ in $\mathbb{R}^d - \{0\}$ and $y_1, \ldots, y_n \in \Omega_\theta^c$ (Evans, 2022). Since $\phi$ is harmonic in $\Omega$, we only have to fit the boundary condition, for instance, in an $L^2$ sense, which amounts at solving

$$\min_{c_1, \ldots, c_n \in \mathbb{R}} \int_{\partial \Omega} \left| \tilde{\phi}(x) - \frac{x_1^2}{2} \right|^2 d\sigma.$$

Since this integral cannot be analytically computed for a general $\Omega$, it is discretized and the resulting least squares problem is solved using `torch.linalg.lstsq`.

There exists several valid choices for the placement of the sources $y_1, \ldots, y_n$. In our case, we decided to do it the following way: draw samples $x_1, \ldots, x_n \in \partial B$, and set $y_i := \phi_\theta(x_i) + \varepsilon n_\theta(x_i)$ for $\varepsilon > 0$. This way, we have the guarantee that $y_i$ lies outside and at distance $\varepsilon$ of $\Omega_\theta$. An alternative and more flexible approach in the case where $\Omega_\theta$ is parametrized by a gauge function Eq. (4) is to take $y_i = \phi_\theta((1 + \varepsilon)x_i)$. This has the advantage that scaling $\Omega_\theta$ leads to the same scaling of the distance from $y_i$ to $\Omega_\theta$.

It is well known that this method is very sensitive to the source placement, and that the conditioning worsens when the number of sources increase. However, this can be mitigated by an *a posteriori* error estimation using the maximum principle: knowing that $\|\phi - \tilde{\phi}\|_\infty := \sup_{\Omega_\theta} |\phi - \tilde{\phi}| = \sup_{\partial \Omega_\theta} |\phi - \tilde{\phi}| = \sup_{x \in \partial \Omega_\theta} \left| \frac{x_1^2}{2} - \tilde{\phi} \right|$, we can estimate the last quantity by sampling a large amount of points on $\partial \Omega_\theta$. If the error is greater than a certain tolerance (taken in our experiments of the order $10^{-4} - 10^{-5}$), we can modify the source placement parameter $\varepsilon$ and increase the number of sources until the tolerance is reached.

**Remark C.1.** *While we decided to treat the case of the Laplace equation for simplicity, this approach can easily be extended to other, more practical settings where we have access to the fundamental solutions of the PDE, like in linearized elasticity (Marin and Lesnic, 2004) for the optimal design of structures or in the Stokes problem (Alves et al., 2021).*

## D  Comparison with PINNs

As it has already been stressed, PINNs are not particularly well suited in general for the computation of PDE-constrained shape optimization. However, there is some very specific problems for which they can easily be applied, namely when the objective function is the *Dirichlet energy* of the shape. This is the type of energy that has been considered in (Bélières Frendo et al., 2025). In this section, we will compare the use of PINNs with the other methods previously introduced on a similar problem for which we know the analytical solution.

Consider the following problem:

$$\max_{\substack{\Omega \in \mathcal{K} \\ \text{Vol}(\Omega) = 1}} T(\Omega) \tag{11}$$

where $T$ is the torsional rigidity defined in Eq. (9). A foundational result in shape optimization, called the Saint–Venant inequality, states that the solution of this problem is the ball (Brasco and De Philippis, 2016).

The Dirichlet energy associated with Eq. (8) is defined as

$$E(\Omega, v) := \frac{1}{2} \int_\Omega |\nabla v|^2 - \int_\Omega v \tag{12}$$

for $v \in H_0^1(\Omega)$. A fundamental property of the Dirichlet energy is that the minimizer $u_\Omega$ of $E(\Omega, \dot{)}$ is the solution of Eq. (8). Using the fact that $\int_\Omega |\nabla u_\Omega|^2 = \int_\Omega u_\Omega = T(\Omega)$, this implies that $E(\Omega, u_\Omega) = -\frac{T(\Omega)}{2}$. Hence:

$$\min_{\substack{\Omega \in \mathcal{K}, v \in H_0^1(\Omega) \\ \text{Vol}(\Omega) = 1}} E(\Omega, v) = \min_{\substack{\Omega \in \mathcal{K} \\ \text{Vol}(\Omega) = 1}} \min_{v \in H_0^1(\Omega)} E(\Omega, v) = \min_{\substack{\Omega \in \mathcal{K} \\ \text{Vol}(\Omega) = 1}} E(\Omega, u_\Omega) = -\frac{1}{2} \max_{\substack{\Omega \in \mathcal{K} \\ \text{Vol}(\Omega) = 1}} T(\Omega),$$

which means that minimizing $E$ in both $\Omega$ and $v$ is actually equivalent to Eq. (11). Using the following proposition, we can further reformulate the problem:

**Proposition D.1.** *Any minimizer $(\Omega, v)$ of*

$$\min_{\substack{\Omega \in \mathcal{K}, v \in H_0^1(\Omega) \\ Vol(\Omega) = 1}} E(\Omega, v) \tag{13}$$

*is a minimizer of*

$$\min_{\Omega \in \mathcal{K}, v \in H_0^1(\Omega)} \frac{E(\Omega, v)}{Vol(\Omega)^{\frac{d+2}{d}}}. \tag{14}$$

*Reciprocally, if $(\Omega, v)$ is a minimizer of Eq. (14) then $\left(\alpha_\Omega \Omega, \alpha_\Omega^2 v(\cdot / \alpha_\Omega)\right)$ where $\alpha_\Omega = Vol(\Omega)^{-1/d}$ is a minimizer of Eq. (13).*

*Proof.* Using the change of variable formula, we can show that for $\alpha > 0$,

$$E\left(\alpha\Omega, \alpha^2 v(\cdot/\alpha)\right) = \alpha^{d+2} E(\Omega, v).$$

The proposition follows. □

Therefore, we can actually drop the volume constraint and instead solve Eq. (14). For a parametrized set $\Omega_\theta$ and $v \in H_0^1(\Omega_\theta)$, we have

$$E(\Omega_\theta, v) = \frac{1}{2} \int_B A_\theta \nabla(v \circ \phi_\theta^{-1}) \cdot \nabla(v \circ \phi_\theta^{-1}) - \int_B (\mathrm{Jac}\phi_\theta)(v \circ \phi^{-1});$$

moreover, minimizing this last expression with respect to $v \in H_0^1(\Omega_\theta)$ is equivalent to minimizing

$$F(\Omega_\theta, v) = \frac{1}{2} \int_B A_\theta \nabla v \cdot \nabla v - \int_B (\mathrm{Jac}\phi_\theta) v.$$

for $v \in H_0^1(B)$. Finally, by taking $v$ to be a neural network $v_\eta$ where $\eta$ are its parameters, we can approximate Eq. (11) by equivalently minimizing the loss

$$L(\theta, \eta) := \frac{F(\Omega_\theta, v_\eta)}{\mathrm{Vol}(\Omega)^{\frac{d+2}{d}}}.$$

**Remark D.1.** *In order to exactly impose the Dirichlet boundary condition on $\partial B$, we chose $v_\eta$ of the form $v_\eta(x) = dist_B(x) MLP_\eta(x)$ where $dist_B$ is the signed distance function of $B$.*

This approach follows the line of the so-called *Deep Ritz Method* (Yu et al., 2018). In opposition to what was previously described in this paper and to classical shape optimization algorithms, one does not need to fully solve the state equation before updating the shape; the state and the shape are jointly optimized, and one can expect $v_\theta$ to be close to the solution of the state equation only at convergence. Since we do not have to solve a PDE at each iteration, one could expect a certain speedup of this method compared to classical ones; it is however not the case, as is shown in the next experiment.

**Evaluation of the method:**   We compare the speed and accuracy of the previously described method with the maximization of $\frac{T(\Omega_\theta)}{\mathrm{Vol}(\Omega_\theta)^{\frac{d+2}{d}}}$ where $T(\Omega_\theta)$ is computed using either the method of fundamental solutions or the mesh free Galerkin method. While the former is expected to be extremely fast and accurate, it doesn't generalize well to other problems with non-constant source term. On the contrary, the latter is expected to be slower due to the fact that it is not taylor to this particular problem; the tradeoff being that it is able to treat a large variety of problems.

Since we know that the optimal shape is the ball, we can measure the discrepency between the shape $\Omega_\theta$ and $B$ using the *isoperimetric deficit* (Fusco et al., 2008), defined as

$$D(\Omega_\theta) = c_d \frac{\mathrm{Per}(\Omega_\theta)}{\mathrm{Vol}(\Omega_\theta)^{\frac{d-1}{d}}} - 1$$

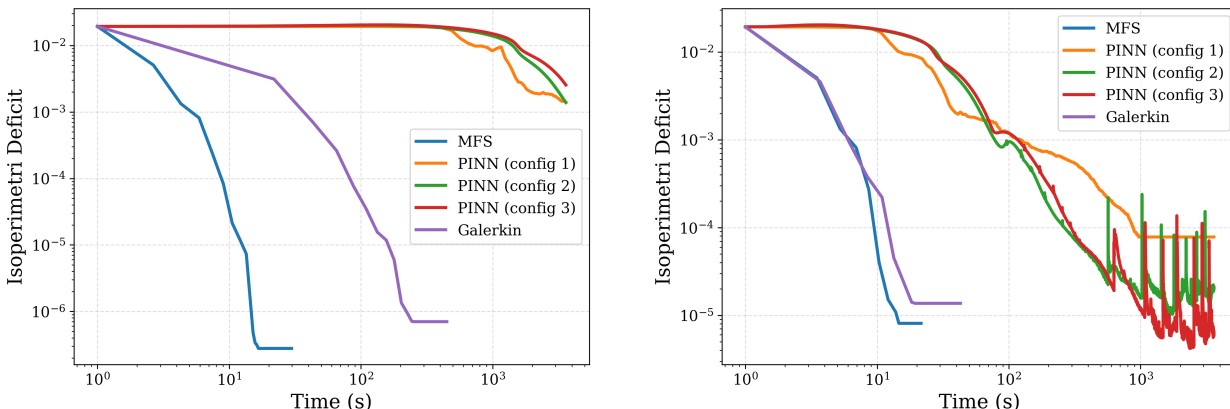

Figure 7: Runtime vs accuracy comparison between classical solvers and PINN-based approaches on CPU (Left) and GPU (Right). Classical methods on CPU achieve higher accuracy at significantly lower computational cost.

with $c_d = (d\mathrm{Vol}(B)^{1/d})^{-1}$. The smallest the isoperimetric deficit, the closest $\Omega_\theta$ is from the ball.

In order to give the best advantage to the PINN approach, we first perform a hyperparameter search for the PINN-based method. The tested parameters are given in Table 2, along with the two best configurations, that we picked in our final experiment. The number of points used for integral evaluation is the same across all experiments ($n = 20000$). The evaluation of the perimeter and volume for the computation of the isoperimetric deficit is made using $n = 100000$ points. On the 72 possible choices of parameters, only 3 reached the precision of $10^{-5}$ eventually; we decided to keep the three of them to compare with the classical methods. These configurations are decribed in Table 2.

Table 2: Hyperparameter search space (left) and the best three configurations that are kept for the final experiment.

| Hyperparameter | Values | Config 1 | Config 2 | Config 3 |
|---|---|---|---|---|
| Neurons per layer | $\{32, 64\}$ | 32 | 32 | 32 |
| Depth | $\{2, 3, 4\}$ | 3 | 3 | 2 |
| Activation Function | { sin, tanh } | tanh | tanh | tanh |
| Optimizer | $\{$Adam, L–BFGS$\}$ | L–BFGS | Adam | Adam |
| Learning rate | $\{10^{-1}, 10^{-2}, 10^{-3}\}$ | $10^{-2}$ | $10^{-2}$ | $10^{-2}$ |

In Fig. 7, we show the comparison between the two classical methods and the three PINN-based ones, both on CPU (AMD Ryzen Pro 7) and on GPU (Nvidia L40). We see that the classical methods consistently outperforms the PINN-based one; interestingly enough, both classical methods ran on CPU are still faster and more accurate than the PINN-based methods ran on GPU. This experiment show that, even in the most favorable setting, the PINN approach is outperformed by the classical methods by several orders of magnitude.

# E   Statistical Analysis

We complement the experiments of Section 4.1 with a statistical evaluation of robustness to noise. For each configuration (shape and noise level), we perform the following procedure. We randomly initialize the sublinear network $p_\theta$, sample $n = 1000$ points $y_i{}_{i=1}^n$, and optimize the loss defined in Section 4.1.

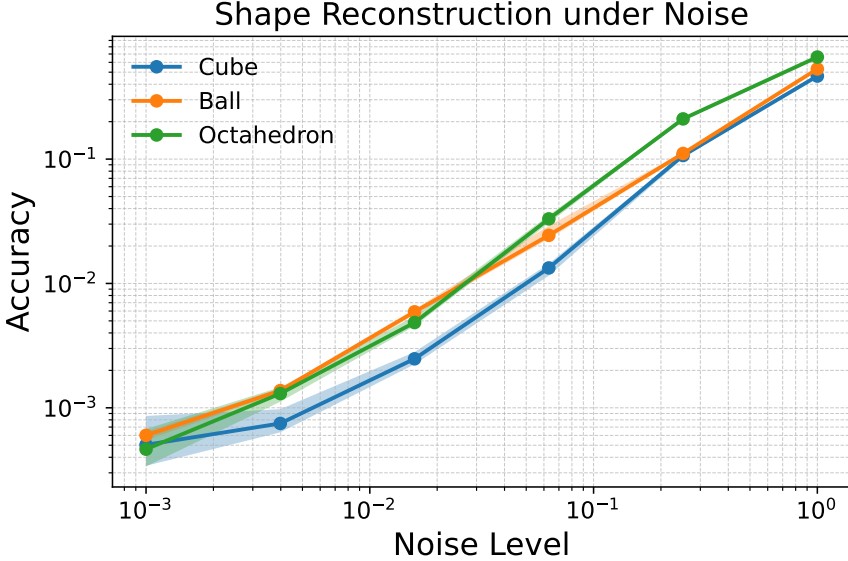

Figure 8: Robustness to noise and network initialization across 100 runs. Median reconstruction error with interquartile range. The method exhibits low variance and stable performance across noise levels.

To quantify reconstruction quality, we measure the $L^2$ discrepancy between the learned function $p_\theta$ and the gauge function of the target shape $g_{\text{target}}$. Since $g_{\text{target}} = 1$ on $\partial\Omega_{\text{target}}$, this reduces to

$$\text{Acc}(\theta) = \|g_{\text{target}} - p_\theta\|_{L^2(\partial\Omega_{\text{target}})} = \|p_\theta - 1\|_{L^2(\partial\Omega_{\text{target}})}$$

We approximate this quantity by sampling $10^5$ points uniformly on $\partial B$ and mapping them via $\phi_{\text{target}}$.

Each experiment is repeated 100 times with independent random initializations and samples. We report the median performance together with the interquartile range (25th–75th percentiles). The aggregated results are shown in Fig. 8.

In a second experiment, we analyze the influence of the number of samples on the reconstruction. We fix the noise level to $\sigma = 0.01$ and use different amount of points $n$ ranging from 10 to $10^4$. The results are reported in Fig. 9.

## F  Integration of the finite element method

Assume that we want to minimize a certain shape function $J$, depending on a PDE that we would like to solve using the finite element method. While possible, it requires to compute certain derivatives manually.

Defining $j(\theta) := J(\Omega_\theta)$, we want to relate the derivatives $\partial_{\theta_k} j(\theta)$ with the shape derivative $dJ(\Omega_\theta).V$. According to (Henrot and Pierre, 2018, Chapter 5, Section 5.9), under some mild regularity assumptions, shape derivatives can be put in the form

$$dJ(\Omega).V = \int_{\partial\Omega} f(x)V(x) \cdot n_\Omega(x) \tag{15}$$

where $f : \partial\Omega \to \mathbb{R}$ depends on $\Omega$. This formula is particularly well suited when one needs to compute $f$ *via* a mesh-based. Now, let $\theta = (\theta_1, \ldots, \theta_n) \in \mathbb{R}^d$ and define $\bar{\theta}(t) := (\theta_1, \ldots, \theta_k + t, \ldots, \theta_n)$. Following (Henrot and Pierre, 2018, Chapter 5, Section 5.2), define

$$\Phi(t)(x) = \phi_{\bar{\theta}(t)} \circ \phi_\theta^{-1}(x)$$

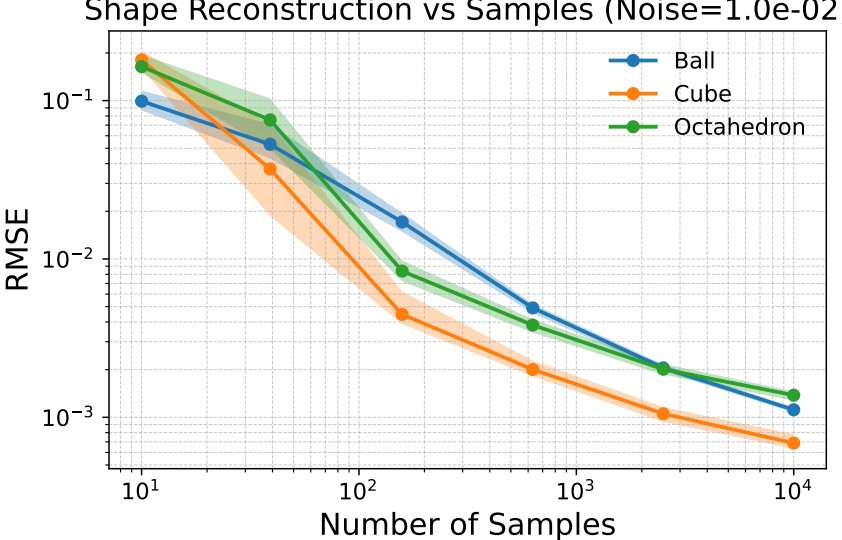

Figure 9: Robustness to amounts of samples and network initialization across 100 runs. Median reconstruction error with interquartile ranges. As expected, the variance is higher when using fewer samples.

for $t \in \mathbb{R}$ and $x \in \Omega_\theta$. We have $\Phi(t)(\Omega_\theta) = \phi_{\bar{\theta}(t)} \circ \phi_\theta^{-1}(\Omega_\theta) = \phi_{\bar{\theta}(t)} \circ \phi_\theta^{-1} \circ \phi_\theta(B) = \Omega_{\bar{\theta}(t)}$. Hence, by letting $V := \Phi'(0)$, we have by definition of the shape derivative that

$$dJ(\Omega_\theta).V = \lim_{t \to 0} \frac{J(\Omega_{\bar{\theta}(t)}) - J(\Omega_\theta)}{t} = \lim_{t \to 0} \frac{j(\bar{\theta}(t)) - j(\theta)}{t} = \partial_{\theta_k} j(\theta).$$

On the other hand,

$$V(x) = \Phi'(0)(x) = \left. \frac{d\phi_{\bar{\theta}(t)} \circ \phi_\theta^{-1}(x)}{dt} \right|_{t=0} = \partial_{\theta_k} \phi_\theta \left( \phi_\theta^{-1}(x) \right),$$

meaning that formally, we have

$$\partial_{\theta_k} j(\theta) = \int_{\partial \Omega_\theta} f(x) \partial_{\theta_k} \phi_\theta \left( \phi_\theta^{-1}(x) \right) \cdot n_{\Omega_\theta}(x) dx. \tag{16}$$

If $\phi_\theta^{-1}$ is easily computable (which is the case, for instance, for Eq. (4) of inverse $\phi_\theta^{-1}(y) = \frac{p_\theta(y)}{\|y\|} y$) and one can precisely evaluate the boundary integral (for instance, if we have a mesh) then this integral can be easily computed. Otherwise, one can formulate it on the reference domain:

$$\partial_{\theta_k} j(\theta) = \int_{\partial B} f(\phi_\theta(x)) \partial_{\theta_k} \phi_\theta \left( \phi_\theta^{-1}(\phi_\theta(x)) \right) \cdot n_{\Omega_\theta}(\phi_\theta(x)) \mathrm{Jac}_{\partial B}(\phi_\theta(x)) dx$$

$$= \int_{\partial B} (f \circ \phi_\theta) \partial_{\theta_k} \phi_\theta \cdot \frac{(D\phi_\theta)^{-T} n_B}{\left| (D\phi_\theta)^{-T} n_B \right|} \mathrm{Jac}_{\partial B}(\phi_\theta)$$

$$= \int_{\partial B} (f \circ \phi_\theta) \partial_{\theta_k} \phi_\theta \cdot (D\phi_\theta)^{-T} n_B \mathrm{Jac}(\phi_\theta)$$

An example of optimal sets using this method can be found in Fig. 10, where we minimize the first 6 Dirichlet eigenvalues under volume and convexity constraint. Specifically, we solve:

$$\min_{\mathrm{Vol}(\Omega)=1} \lambda_k(\Omega) \tag{17}$$

where $\Omega \in \mathcal{K}$ and $\lambda_k(\Omega)$ is the $k$-th Dirichlet eigenvalue, i.e. it solves

$$\begin{cases} -\Delta u = \lambda_k(\Omega)u & \text{in } \Omega \\ u = 0 & \text{on } \partial\Omega \end{cases} \tag{18}$$

for some $u \in H_0^1(\Omega)$. This particular problem has already been considered in (Antunes and Bogosel, 2022).

In order to compute the derivatives Eq. (16) of $\lambda_k$, we created a mesh of the domain $\Omega_\theta$ using `gmsh` (Geuzaine and Remacle, 2009), by giving it a sequence of points $y_i = \phi_\theta(x_i)$ where $x_i = (\cos(2i\pi/n), \sin(2i\pi/n))^T$, $1 \leq i < n$. An eigenpair $\lambda(\Omega_\theta), u(\Omega_\theta)$ is computed *via* finite elements using `scikit-fem` (Gustafsson and Mcbain, 2020). Using the Hadamard expression for the shape derivative $d\lambda(\Omega).V = \int_{\partial\Omega} |\nabla u|^2 (V \cdot n)$, we deduce that

$$\partial_{\theta_k} j(\theta) = \int_{\partial\Omega_\theta} |\nabla u(x)|^2 \partial_{\theta_k}\phi_\theta\left(\phi_\theta^{-1}(x)\right) \cdot n_{\Omega_\theta}(x)dx.$$

The differential quantity $\nabla u(x)$ is computed on the finite element space, while $\partial_{\theta_k}\phi_\theta$ is computed using AD. The integral is computed on the boundary mesh by `scikit-fem`. All of this process is wrapped in `torch.autograd.Function` in order to seamlessly integrate it into PyTorch's AD.

You can see the six optimal shapes found by the algorithm in Fig. 10 along with the corresponding eigenvalue and the relative error $E_{\mathrm{rel}} := \frac{\lambda_k - \lambda_k^*}{\lambda_k^*}$ where $\lambda_k^*$ is either the optimal value obtained numerically in (Antunes and Bogosel, 2022) or the analytical one for $k = 1$ and $k = 3$ where the optimal shape is known to be the ball. We see that the shapes are in good agreement with the previous results, and that the relative error is of order $-3$ to $-4$.

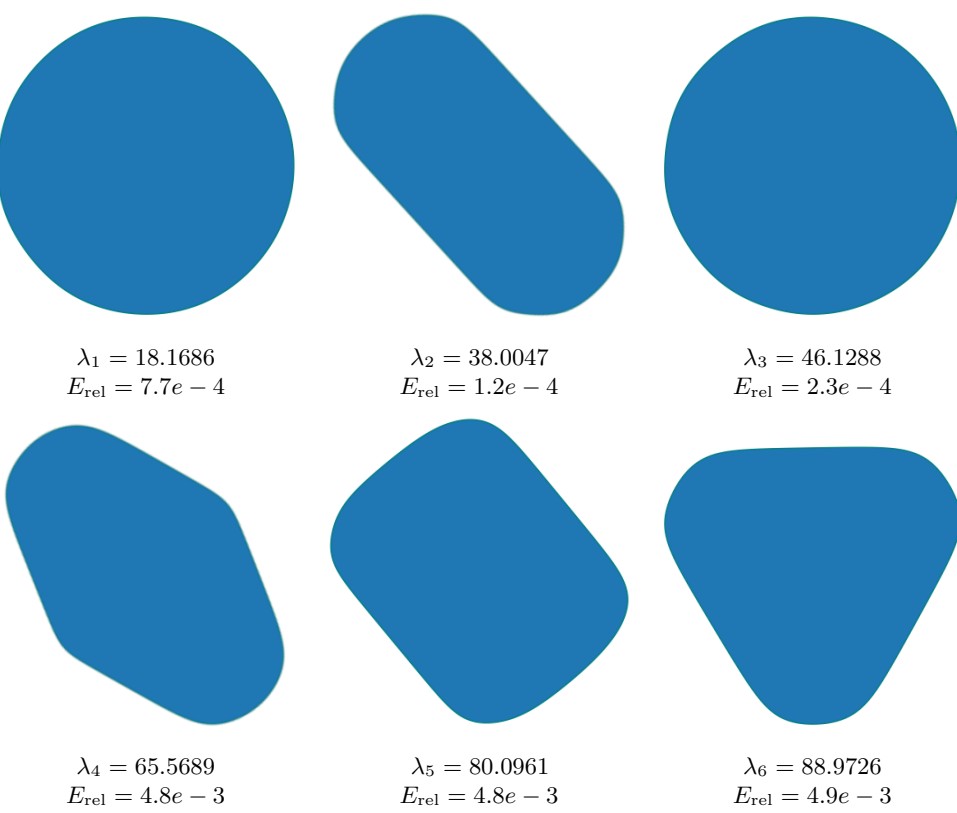

$\lambda_1 = 18.1686$
$E_{\mathrm{rel}} = 7.7e - 4$

$\lambda_2 = 38.0047$
$E_{\mathrm{rel}} = 1.2e - 4$

$\lambda_3 = 46.1288$
$E_{\mathrm{rel}} = 2.3e - 4$

$\lambda_4 = 65.5689$
$E_{\mathrm{rel}} = 4.8e - 3$

$\lambda_5 = 80.0961$
$E_{\mathrm{rel}} = 4.8e - 3$

$\lambda_6 = 88.9726$
$E_{\mathrm{rel}} = 4.9e - 3$

Figure 10: Optimal shapes for the first six Dirichlet eigenvalues

