# OpenReview forum: "Parametrizing Convex Sets Using Sublinear Neural Networks"
_TMLR — Under review for TMLR_

### Review · Reviewer_uu1E · 2026-06-30

**Summary Of Contributions:**

This paper proposes a neural network parameterization of convex sets and proves a universal approximation theorem for such convex sets. The parameterization is done through gauge or support functions of convex sets. The theory is thorough and explained in detail. Authors also show how the results can be used to compute different shape quantities, such as integrals, curvitures, and PDE solutions. The experiments are relevant and varied.

**Audience:**

Yes

**Audience Explanation:**

Anybody working on shape analysis for convex shapes will be interested in the findings of this paper.

**Broader Impact Concerns:**

No concerns

**Claims And Evidence:**

Yes

**Claims Explanation:**

All theorems are proved and the proofs are correct (up to minor issues remarked below).

**Requested Changes:**

- Several of the proofs rely heavily on theorems that are never stated in the paper. For example (Schneider, Theorem 1.8.19, Lemma 1.8.14). Presenting these in the appendix would make the paper much easier to read and self-contained. [strengthening]
- In places, the notation introduced in the appendix is critical to understand the paper. The main paper should at least introduce these. For example, mention that $f^{\ast}$ is the complex conjugate. It's fine to have the definition in the appendix, but at least mention what the object is refering to. [critical]
- The related literature is not sufficiently taken into account. Previous work has used sublinear functions to learn the support of convex sets, e.g.  S. Haddad and A. Halder, "Convex and Nonconvex Sublinear Regression with Application to Data-driven Learning of Reach Sets," 2023 American Control Conference (ACC), San Diego, CA, USA, 2023, pp. 4581-4586, doi: 10.23919/ACC55779.2023.10156204. The paper needs to clearly state its contributions compared to the previous work. [critical]
- In the proofs of Propositions 2.1 and 2.2, the $K$ should probably be $\phi(B)$. [critical]
- The proof of Proposition 2.4 does not show that the infimum is taken over a non-empty set, which is necessary for it to be bounded above. [critical]
- The invariances using frame averaging are ok, but a more interesting version would directly restrict $W$. If the group are rotations $R$, it is easy to see that for invariance to hold, we need $W^T(R-I)=0$. This restriction could be directly included in the neural network. [strengthening]
- It could at times be more clear that gauge and support individually produce the results and not both are necessary. For example Theorem 2.1 can be misunderstood to mean that $K^{NN}$ is the union of all convex sets that come from gauge functions and the convex sets parameterized from support functions. Also, the start of Section 3.4 could be moved to the introduction to make this clear earlier. [strengthening]
- "Lawrence C Evans. Measure theory and fine properties of functions. 2025." should probably be "Lawrence C Evans, Ronald F Gariepy. Measure theory and fine properties of functions. 2015." [critical]

---

> ### Author Response · Authors · 2026-07-14
>
> I would like to thank the reviewer for the careful and insightful review. I think that the comments greatly helped to improve the clarity of the paper. Thank you especially for pointing out the incompleteness of Proposition 2.4. All the requested changes have been taken into account. For ease of reference, I have reproduced your original comments and provided my responses beneath them in a tabulated format.
>
>
> Several of the proofs rely heavily on theorems that are never stated in the paper. For example (Schneider, Theorem 1.8.19, Lemma 1.8.14). Presenting these in the appendix would make the paper much easier to read and self-contained. [strengthening]
>
>     Most of the references to Schneider have been added in the Appendix.
>
>
> In places, the notation introduced in the appendix is critical to understand the paper. The main paper should at least introduce these. For example, mention that is the complex conjugate. It's fine to have the definition in the appendix, but at least mention what the object is refering to. [critical]
>
>     This has been fixed
>
>
> The related literature is not sufficiently taken into account. Previous work has used sublinear functions to learn the support of convex sets, e.g. S. Haddad and A. Halder, "Convex and Nonconvex Sublinear Regression with Application to Data-driven Learning of Reach Sets," 2023 American Control Conference (ACC), San Diego, CA, USA, 2023, pp. 4581-4586, doi: 10.23919/ACC55779.2023.10156204. The paper needs to clearly state its contributions compared to the previous work. [critical]
>
>     The reference has been added in section 2.2, before stating that we need in our case a smooth parametrization for computations of higher-order quantities. Another reference (which appeared after the first version of the present paper) has also been added at the end of section 2.2 (Theo X. Olausson, João Monteiro, Michal Klein, and Marco Cuturi. Amortized maximum inner productsearch with learned support functions, 2026)
>
>
> In the proofs of Propositions 2.1 and 2.2, the K should probably be \phi(B). [critical]
>
>     In Proposition 2.1 and Proposition 2.2, I have replaced the K by \Omega. The unnecessary notation $\tilde \Omega$ has also been replaced by $\phi(B)$ for clarity purposes.
>
>
> The proof of Proposition 2.4 does not show that the infimum is taken over a non-empty set, which is necessary for it to be bounded above. [critical]
>
>     The proof was indeed incomplete, thank you for pointing it out. I added the assumption that W has full rank, which is always the case in practice.
>
>
> The invariances using frame averaging are ok, but a more interesting version would directly restrict W. If the group are rotations R, it is easy to see that for invariance to hold, we need W^T(R-I)=0. This restriction could be directly included in the neural network. [strengthening]
>
>     I agree that this constraint would allow invariance to hold; however, unless I am mistaken, it is way too restrictive. Indeed, it reduces at asking that RW = W for all R in the group, or equivalently that every column of W is invariant under R; in 2D for instance, this can not be veryfied unless the group is reduced to the identity or W = 0.
>
>
> It could at times be more clear that gauge and support individually produce the results and not both are necessary. For example Theorem 2.1 can be misunderstood to mean that is the union of all convex sets that come from gauge functions and the convex sets parameterized from support functions. Also, the start of Section 3.4 could be moved to the introduction to make this clear earlier. [strengthening]
>
>     Theorem 2.1 has been modified: it is now quantitative, and split into Theorem 2.1 and 2.2 for clarity. I defined two different set $\mathcal K^s$ and $\mathcal K^g$ for the gauge and support parametrization.
>
>     The start of Section 3.4 has been moved to the end of the intorduction and slightly adapted.
>
> "Lawrence C Evans. Measure theory and fine properties of functions. 2025." should probably be "Lawrence C Evans, Ronald F Gariepy. Measure theory and fine properties of functions. 2015." [critical]
>
>     That has been fixed

---

> > ### Comment · Reviewer_uu1E · 2026-07-16
> >
> > The paper has been substantially improved according to the reviewer comments.
> >
> > I still think that the previous work should go into the related works as well, to provide appropriate background.
> >
> > And I agree with the other reviewers that the fact that the shallow architecture cannot be extended to larger depth is a major drawback of the method. This should receive more attention at least in the discussion.

---

### Review · Reviewer_E6Lh · 2026-07-01

**Summary Of Contributions:**

Consider the set K of convex subsets of the Euclidean vector space ℝᵈ. This
paper discusses the support and gauge functions as two representations of
elements of K. By a particular neural approximation of these, they yield
a parameterized subset of K, for which they provide a universal approximation
theorem. Different learning tasks in this setting form experimental evaluations
of this construction.


Strengths:

 - Elegant idea from classic tools from convex geometry, yielding exact
   convexity by construction. Nice connection between support and gauge
   function and neural parameterization

 - Universal approximation theorem

 - The experimental evaluations are interesting from a pure mathematical
   perspective

Weaknesses:

- Introduction and motivation needs substantial improvement. Right now it
  directly jumps to prior and related work, without making clear why the reader
  should be interested in this problem at all. In §1 often the context is
  missing. The motivation is missing entirely, making it hard for the reader to
  put this paper into the right perspective.

- Much of the theoretical development relies on standard constructions from
  convex geometry (e.g., support/gauge representations, polytope
  approximations, duality arguments). While the combination of these tools is
  elegant, the theoretical novelty beyond their integration into the proposed
  framework is less clear.

- The presented neural architecture is quite shallow, i.e., a single linear layer
  with a LogSumExp pooling.  Speaking of "sublinear neural networks" sounds
  a bit like overselling.

- The experiments are either very academic or rather look like toy examples.

**Additional Comments:**

The manuscript would benefit from careful proofreading.

In (2), if you have θ on the left of an equation then we would also like to see
it on the right.

In §2.1 you speak of convex bodies in ℝᵈ, but you need to give this set
a mathematical structure. Later on you rely on ℝᵈ forming an Euclidean vector
space.

In Table 2, each config has layer size 32? All layers had to have the same
size? Why? And maybe the value range for the hyperparameter tuning was bad?
What was the rationale behind the learning rate and activation function value
range here? And again all layers have this activation function?

In §4.5 you speak of ℝᵈ when you mean ℝ².

**Audience:**

Yes

**Audience Explanation:**

Convexity is a core concept in geometry with many applications in STEM. The
paper could improve the presentation in this regard; see my comment on the
motivation section.

**Claims And Evidence:**

No

**Claims Explanation:**

- The representation efficiency is not analyzed, e.g., the effect on the size
  of the layer or the effect of the β smoothing parameter.

- It is unclear how much power of the method to attribute to the neural
  architecture or to the support/gauge function representation. An ablation
  study on the size of network would be interesting.

- The universal approximation theorem is good to have, but it is a rather
  immediate result from standard tools of convex geometry. However, what would
  be very interesting is a statement that would give us bounds on the size $m$
  of the neural layer. And if it is to hard in theory then an experiment would
  be interesting.

- Remark 2.2 states that extending the construction to deeper architectures is
  non-trivial. However, it remains unclear what fundamental obstacle prevents
  such an extension. Even if exact sublinearity were lost, it would be useful
  to better understand whether deeper architectures could provide a more
  efficient representation class. Some discussion or empirical exploration of
  this design space would strengthen the paper.

**Requested Changes:**

- Comparison to baselines would be indicated. None of the experiments explains
  how the proposed method compares to the state of the art.

- A quantified evaluation of the experiments would be indicated. For instance
  in §4.1, how far are the reconstructions from the ground truth, or the convex
  hull of the samples, (or some baseline).

- Ablation studies, so we learn where the power of this proposed method stems
  from.

---

> ### Author Response · Authors · 2026-07-14
>
> Thank you very much for your thorough and thoughtful review. Your comments have greatly helped improving the quality and clarity of the manuscript. Please find below my detailed responses to each of your comments. For ease of reference, I have reproduced your original comments and provided my responses beneath them in a tabulated format. My answer is split into two comments since the answer exceeded the character limit.
>
> Introduction and motivation needs substantial improvement. Right now it directly jumps to prior and related work, without making clear why the reader should be interested in this problem at all. In §1 often the context is missing. The motivation is missing entirely, making it hard for the reader to put this paper into the right perspective.
>
>     I added two introductory paragraphs. I hope that this puts enough context for any reader that is not familiar with convex shape analysis.
>
>
> Much of the theoretical development relies on standard constructions from convex geometry (e.g., support/gauge representations, polytope approximations, duality arguments). While the combination of these tools is elegant, the theoretical novelty beyond their integration into the proposed framework is less clear.
>
>     The novelty is probably more numerical than theoretical: namely, it enforces exact convexity of the shapes without any constraint, while classical approaches in convex shape optimization rely on approximate convexity enforcement (that can break during the optimization process) or only works in dimension 2. Moreover, I strengthened theorem 2.1 to make it quantitative; while it is still based on classical convex-geometric arguments, I hope that it makes the paper of greater interest.
>
>
> The presented neural architecture is quite shallow, i.e., a single linear layer with a LogSumExp pooling. Speaking of "sublinear neural networks" sounds a bit like overselling.
>
>     I did not mean to coin a term, I am just referring to the fact that the neural networks that I consider in this work are sublinear. It would be helpful if you could suggest a better way to phrase it.
>
>
> The experiments are either very academic or rather look like toy examples.
>
>     I agree. I added in the introduction that numerical convex shape optimization can highly benefit to the mathematical intuition, and most shape optimization problems of this kind are closer to the theory than to applications. I would be glad if you could suggest some relevant, less simplistic experiments that could be conceived and run in a decent amount of time.
>
>
> The representation efficiency is not analyzed, e.g., the effect on the size of the layer or the effect of the β smoothing parameter.
>
>     This is now hopefully solved: I split the Theorem 2.1 in two (Theorem 2.1 and Theorem 2.2) and made it quantitative, i.e. giving the amount of neurons necessary to get an error below $\epsilon$.
>
>
> It is unclear how much power of the method to attribute to the neural architecture or to the support/gauge function representation. An ablation study on the size of network would be interesting.
>
>     I believe that the new, quantitative UATs now have more power than an ablation study; I would however be glad to make one if you still deem it necessary.
>
>
> The universal approximation theorem is good to have, but it is a rather immediate result from standard tools of convex geometry. However, what would be very interesting is a statement that would give us bounds on the size of the neural layer. And if it is to hard in theory then an experiment would be interesting.
>
>     As stated above, the UATs are now quantitative.

---

> ### Author Response · Authors · 2026-07-14
> **Asnwer to the review: second part.**
>
> Remark 2.2 states that extending the construction to deeper architectures is non-trivial. However, it remains unclear what fundamental obstacle prevents such an extension. Even if exact sublinearity were lost, it would be useful to better understand whether deeper architectures could provide a more efficient representation class. Some discussion or empirical exploration of this design space would strengthen the paper.
>
>     As stated in Remark 2.2, the interaction between convex conjugation and function composition is not trivial, the conjugation transforming the composition into infimal postcomposition. Analyzing the sublinearity of a deeper neural network would necessitate to consider postcompositions of postcompositions with non-linear function, which quickly become untractable in our setting. Moreover, some experiments that I performed by composing such sublinear layers shows that it in fact breaks the convexity of the target shape in practice.
>
>     Another work published after this one's submission (Amortized Maximum Inner Product Search with Learned Support Functions, Theo X. Olausson, João Monteiro, Michal Klein, Marco Cuturi) actually considers homogeneous extensions of networks that are not sublinear; however, in the case of the present work, the aim is precisely to enforce exact convexity, as it generally missing in classical shape optimization methods. Moreover, certain problems considered here (like the Minkovski problem) requires exact convexity to be well posed.
>
>
> Comparison to baselines would be indicated. None of the experiments explains how the proposed method compares to the state of the art.
>
>     As it is stated in the second paragraph or Section 4: " A direct quantitative comparison with classical methods is not possible in practice, as no publicly available implementations exist in a form that would allow a consistent and reproducible evaluation". If you have knowledge of such methods that have a publicly available implementation, I would add a comparison in the present paper.
>
>
> A quantified evaluation of the experiments would be indicated. For instance in §4.1, how far are the reconstructions from the ground truth, or the convex hull of the samples, (or some baseline).
>
>     If I understand the request correctly, this is done in Appendix D.
>
>
> Ablation studies, so we learn where the power of this proposed method stems from.
>
>     Please suggest ablation studies that you think would be relevant.
>
>
> The manuscript would benefit from careful proofreading.
>
>     I hope that the manuscript is now better in term of typos/discrepancies.
>
>
> In (2), if you have θ on the left of an equation then we would also like to see it on the right.
>
>     Thank you for noticing. I added explicitly that the parameters $\theta$ are $W$ and $\beta$.
>
>
> In §2.1 you speak of convex bodies in ℝᵈ, but you need to give this set a mathematical structure. Later on you rely on ℝᵈ forming an Euclidean vector space.
>
>     I added a short sentence at the begining of the section.
>
> In Table 2, each config has layer size 32? All layers had to have the same size? Why? And maybe the value range for the hyperparameter tuning was bad? What was the rationale behind the learning rate and activation function value range here? And again all layers have this activation function?
>
>     Each of the *best* config indeed has size 32. The layers do not need to have the same size, nor the same activation function; however, this is pretty standard in the PINN community and allowing for different sizes would further multiply the amount of tests to perform in the hyperparameters seach, which already takes a significant amount of time (several hours to days).
>
>     The choice of values for the learning rates is consistent with the literature (see eg https://arxiv.org/pdf/2408.00573, or the code of the original PINNs paper by Raissi), as well as the activation functions which are among the most popular choices in the community (one could also have considered ReLU2 or ReLU3 activation functions too).
>
>     If the poor performance of PINNs compared to the classical methods makes you think that the range of hyperparamteters was bad, I refer you to https://arxiv.org/abs/2302.04107 or https://www.nature.com/articles/s42256-024-00897-5. If there is another reason, I would be happy to hear it.
>
> In §4.5 you speak of ℝᵈ when you mean ℝ².
>
>     Thank you for noticing.

---

> > ### Comment · Reviewer_E6Lh · 2026-07-15
> >
> > We thank the authors for their substantive improvements and the reply to the
> > reviewer's feedback. Here is an update of the reviewers view:
> >
> > # Improved
> >
> > - Introduction has been improved
> >
> > - Approximation rates are an important improvement and add value to the paper.
> >   It also addresses a couple of further questions en passant.
> >
> > # Somewhat resolved
> >
> > - Weak baselines: The authors argue that code is not available for
> >   a comparison. Although it would have strengthened the evaluation
> >   significantly, we acknowledge this limitation.
> >
> > - Experiments (academic or toy example): We acknowledge the authors honesty in
> >   the answer.
> >
> > # What was not improved
> >
> > - Shallow architecture: A major aspect of neural-network methods in machine
> >   learning is their architectural flexibility, including the choice of depth,
> >   activation functions, layer widths, and specialized structures. This
> >   flexibility spans MLPs, U-net architectures, recurrent architectures, and so
> >   on. Fundamentally having a hard restriction to one single linear layer with
> >   a fixed aggregation is quite contrary to what the scientific community would
> >   expect when positioning it as "sublinear neural networks". From an ML
> >   perspective, the proposed model is closer to a specialized sublinear layer.
> >   We would encourage the authors to clarify the positioning.
> >
> > - Missing ablation study. The authors argue that the quantitative approximation
> >   theorem is more powerful than an ablation study. While the authors have some
> >   point here, we would still argue that an approximation theorem and ablation
> >   studies serve different purposes. In particular, Thm-2.3, which is a valuable
> >   theoretical addition, only provides existence of the relevant constants,
> >   i.e., it establishes asymptotic bounds. An ablation study would provide
> >   practical insight on the concrete behavior and sensitivity, the concrete
> >   impact and scaling behavior of different parameters.
> >
> > # Some further details
> >
> > - Related work: Gosh and Kumar [1] give an overview of different support
> >   function representations, e.g., the Fourier series expansion. This appears to
> >   be related to your work. The original work is Groemer, Fourier series and
> >   spherical harmonics in convexity, 1993. This again could hint to future work
> >   of your submission?
> >
> > - Limitation of intersection of half-spaces for smooth geometries: The
> >   half-space intersection can be phrased by means of a max/min expression of
> >   the inner product with the orthogonal supporting vectors. By replacing this
> >   with an LSE-based formulation, also these can be made smooth, just like you
> >   turned max_i W^T⋅x into a smooth expression, right?
> >
> >
> > [1] Gosh and Kumar, Support Function Representation of Convex Bodies, Its
> > Application in Geometric Computing, and Some Related Representations, COMPUTER
> > VISION AND IMAGE UNDERSTANDING Vol. 72, No. 3, December, pp. 379–403, 1998

---

### Review · Reviewer_Hq8A · 2026-07-02

**Summary Of Contributions:**

This paper describes a neural network functional approximation of convex gauge functions, which in turn is used to optimize and approximate shapes for various PDE applications.

Strengths: The functional NN proposed is interesting, and the authors went into depth to try to characterize it well

Weaknesses: The paper is not written in a way that clearly conveys what contributions it is giving to the ML community. This is from 1. theorems that are not rigorously stated and proofs that are not easy to follow and often have big holes in them, 2. putting way too many details in the appendix rather than the real paper, 3. not clearly describing the road map, objectives, and toolsets with regards to the PDE parts.

I think it is possible for this paper to be revised and eventually be accepted, but in my view a heavy rewrite is necessary. (The science and math may actually be sufficient though; it's just not verifiable in the current state.)

**Audience:**

Yes

**Audience Explanation:**

I think using NNs for scientific modeling is of interest to the community.

**Claims And Evidence:**

No

**Claims Explanation:**

I have written in more detail in "requested changes". Basically, the proofs need to be cleaned up.

**Requested Changes:**

**Citations**

The paper seems to mainly cite Schneider 2013, but I believe many of these results precede this, including Rockafellar (Convex analysis), Freund (Dual gauge programs, with applications to quadratic programming and the minimum-norm problem, '87), M. P. Friedlander, I. Macêdo, T. K. Pong, "Gauge optimization and duality," (2014). I would recommend a deeper literature study so that the context of this work is clear.

**Proof of proposition 2.1** (relatively minor concerns)

 - Looks correct, but what is $K$? It's never defined. $\mathcal K$ is a set of sets so it cannot be the same thing.
 - We are assuming $\|\cdot\|$ is 2-norm right? Otherwise it may not be differentiable even outside of $x = 0$.ok
 - The last sentence of the proof is true but doesn't seem like an implication from the previous sentences -- it is more a result that $g(B) = \mathbf{conv}(\mathbf{cl}( \Omega))$.

**Proof of proposition 2.2** (potentially major concerns)
 - (minor) there's a $y$ which is not math formatted
 - Again, what is $K$?
 - "Hence $\tilde \Omega = \Omega$ meaning that $\phi(B)$ is convex." again I follow all the arguments but I don't see how it leads to this implication. I can see it is true though, since the polar of the polar of a convex set is convex, so if the two sets are equal then the original set is convex. I think that's the argument you're using? But it feels roundabout.
 - It might help to cite something to be able to say that $\nabla h_\Omega(x) \in \partial \Omega$. By the way, $\partial \Omega$ is never defined.
 - "However, since $h > 0$ we have that $0\in \mathbf{int}(\Omega)$" It is never stated anywhere that $h > 0$.
 - "Hence any half line originating at 0 must intersect $\partial \Omega$ exactly once" I agree but I don't see how that is proven here
 - "Using that $\nabla h$ is a homeomorphism, this means that $x_1/\|x_1\|=x_2/\|x_2\|$ ..." Again, agree with the statement but don't see why.

**Theorem 2.1** (major concerns)
 - The statement of Theorem 2.1 doesn't make any sense. I think you mean the set of $\phi_\theta(B)$, but then how does $\theta$ connect with $\beta$, $W$, or $m$? Later, it is stated that $\theta = (\beta,W)$ but that really should be presented before the proof.
 - The result about $d_H(K,P) \leq \epsilon/2$, is this for all $\epsilon > 0$? Is there a dependence on $m$? dimensionality? Sure, any convex body can be approximated in the limit by a polytope of infinite vertices, but that isn't clearly conveyed, if that is the point here.
 - I generally would avoid using phrases like "It is well known that" in a paper like this. I do know because I spent a long time studying this subject, but for a general purpose machine learning focused audience, for any fact that is critical to a proof, I'd recommend either a citation or a short intuitive explanation.
 - I basically cannot really check the proof of this theorem because the theorem statement is not clearly stated. Since $W$ is not introduced in the definition of $p_\theta$, I'm not sure which polytope I should be comparing to which set, and why this means a function class is dense in the class of all convex (presumably closed and lower semicontinuous) functions. I can see where the argument is going and feel some reasonable statement is possible here, but as of this form I don't think it passes muster.

**Proposition 2.5** (major concerns)
 - in the function $p_\theta^T(x):=\frac{1}{|G|}\sum_{g\in G}p_\theta(x)$, what is $g$ actually doing? Do you mean $p_\theta^T(x):=\frac{1}{|G|}\sum_{g\in G}p_\theta(g.x)$?
 - I'm not sure what $\tilde g$ is, how it factored in
 - I believe there is a typo in Case 2, and it's affecting what the statement s trying to say. It is not clear to me if this proof goes through.

**Section 3 and 4** (major concerns)
 - Unlike section 2 which was pretty step by step, sections 3 and 4 read more like a list of known facts patched together. It is very hard to figure out what was the goal and how the neural network $\phi_\theta$ comes into play. I think given that this is an ML journal and not a PDE journal, a clearer roadmap is needed, and to actually show what role that $p_\theta$, both in training and modeling, is giving.
- I was not clear as to how $p_\theta$ is being trained. Where is the objective function? Is it constrained? I saw that L-BFGS is used, but to optimize what exactly?
- If numerics is an important contribution, there should be more baselines to compare against.

**Overall**

 - It's confusing to use $\phi$ for two separate functions. Suggest splitting them into two symbols.

---

> ### Author Response · Authors · 2026-07-14
>
> Thank you for the very thorough review, that helped greatly improve this manuscript. Please find the detailed, point-by-point answers to each of your questions hereafter. I may need some more clarification on certain aspects. I have reproduced your original comments and provided my responses beneath them in a tabulated format. My answer is split into two comments since the answer exceeded the character limit.
>
> ---
>
> The paper seems to mainly cite Schneider 2013, but I believe many of these results precede this, including Rockafellar (Convex analysis), Freund (Dual gauge programs, with applications to quadratic programming and the minimum-norm problem, '87), M. P. Friedlander, I. Macêdo, T. K. Pong, "Gauge optimization and duality," (2014). I would recommend a deeper literature study so that the context of this work is clear.
>
>     I did not mean to refer to the papers where the stated results originally appeared, since those are all classical results; moreover, Schneider is a  very classical reference, which have the advantage of containing all of the relevant results used in this paper. I however added a citation to Rockafellar, thank you for pointing this out.
>
>
> Proof of proposition 2.1 (relatively minor concerns)
>
> Looks correct, but what is $K$ ? It's never defined. $\mathcal K$ is a set of sets so it cannot be the same thing.
>
>     Indeed, it was supposed to be $\Omega$. Thank for pointing it out.
>
> We are assuming $|.|$ is 2-norm right? Otherwise it may not be differentiable even outside of $0$.ok
>
>     I added a sentence to make clear that $\|.\|$ is the Euclidean norm at the beginning of subsection 2.1
>
> The last sentence of the proof is true but doesn't seem like an implication from the previous sentences -- it is more a result that $g(B) = cov(cl(\Omega))$.
>
>     This is very true: we showed that $\phi(B) = \Omega$, and $\Omega$ is obviously convex (as the sublevel-set of a convex function) and closed (as the preimage of a closed set by a continuous function). I added a sentence to make this clear at the beginning of the proof.
>
>
>
> Proof of proposition 2.2 (potentially major concerns)
>
> (minor) there's a $y$ which is not math formatted
>
>     Thank you for noticing, this has been fixed.
>
>
> Again, what is $K$ ?
>
>     Same typo as before, this has been fixed.
>
> "Hence $\tilde \Omega = \Omega$ meaning that $\phi(B)$ is convex." again I follow all the arguments but I don't see how it leads to this implication. I can see it is true though, since the polar of the polar of a convex set is convex, so if the two sets are equal then the original set is convex. I think that's the argument you're using? But it feels roundabout.
>
>     First note that I removed the unnecessary $\tilde \Omega$ notation and directly used $\phi(B)$. I do not use this argument, I just proceed by a standard double inclusion. I added a short sentence " and $\Om \subset \phi(B)$" to hopefully make this clearer.
>
>
> It might help to cite something to be able to say that $\nabla h_\Omega(x) \in \partial \Omega$. By the way $\partial \Omega$, is never defined.
>
>     I actually cited Schneider, Corollary 1.7.3 (now Proposition A.4) just before Proposition 2.2 to say that $\nabla h_\Om : \S^{n-1} \to \partial \Om$ is a homeomorphism. I defined $\partial \Omega$ directly after this, thank you for pointing it out.
>
>
> "However, since $h > 0$ we have that $0 \in int(\Omega)$" It is never stated anywhere that $h > 0$.
>
>     It is actually stated in the proposition : "Let h be a positive and sublinear function..."
>
>
> "Hence any half line originating at 0 must intersect exactly once" I agree but I don't see how that is proven here.
>
>     It follows by very elementary properties of convex sets. I did not show it to avoid making the proof too packed, but I can definitely add a few line on this if you deem it necessary.
>
>
> "Using that $\nabla h$ is a homeomorphism, this means that ..." Again, agree with the statement but don't see why.
>
>     It was indeed unclear, I added some steps to better convey the argument.
>
>
> Theorem 2.1 (major concerns)
>
> The statement of Theorem 2.1 doesn't make any sense. I think you mean the set of $\phi_\theta(B)$, but then how does connect with $\beta$, $W$, or $m$? Later, it is stated that $\theta = (\beta, W)$ but that really should be presented before the proof.
>
>     Indeed, it has now been introduced before (just after equation (3)). Also, $\mathcal K^\text{NN}$ has been replaced by $\mathcal K^s$ and $\mathcal K^g$, and the theorem has been split into Theorem 2.1 and 2.2. $\theta$ has been explicitly defined as $(\beta, W)$ on the first line of page 5.

---

> ### Author Response · Authors · 2026-07-14
> **Second part**
>
> The result about $d_H(K,P) < \epsilon/2$, is this for all $\epsilon>0$? Is there a dependence on $m$? dimensionality? Sure, any convex body can be approximated in the limit by a polytope of infinite vertices, but that isn't clearly conveyed, if that is the point here.
> I generally would avoid using phrases like "It is well known that" in a paper like this. I do know because I spent a long time studying this subject, but for a general purpose machine learning focused audience, for any fact that is critical to a proof, I'd recommend either a citation or a short intuitive explanation.
> I basically cannot really check the proof of this theorem because the theorem statement is not clearly stated. Since is not introduced in the definition of , I'm not sure which polytope I should be comparing to which set, and why this means a function class is dense in the class of all convex (presumably closed and lower semicontinuous) functions. I can see where the argument is going and feel some reasonable statement is possible here, but as of this form I don't think it passes muster.
>
>     Theorem 2.1 has been substantially transformed: i split it into Theorem 2.1 and Theorem 2.2 to make clear which parametrization (support or gauge) we are using in each case. Moreover, I improved the result by giving rates of convergence, i.e. the number $m$ of neurons necessary to approximate a convex set to order $\epsilon$. I hope that this would make the results clearer and more powerful, overall strengthening the paper.
>
>
> Proposition 2.5 (major concerns)
>
> in the function $p_\theta^G(x) := \frac{1}{|G|} \sum_{g \in G} p_\theta(x)$, what is $g$ actually doing? Do you mean $p_\theta^G(x) := \frac{1}{|G|} \sum_{g \in G} p_\theta(g.x)$?
>
>     Indeed, thank you for noticing, this has been fixed.
>
>
> I'm not sure what $\tilde g$ is, how it factored in
>
>     $\tilde g$ is just a dummy variable of the sum over $G$. It factored in by using the change of variable $\tilde g \xleftarrow{} \tilge g g$. I added a step, I hope it makes it clearer.
>
>
> I believe there is a typo in Case 2, and it's affecting what the statement s trying to say. It is not clear to me if this proof goes through.
>
>     Indeed, the typo is corrected. If the proof needs further clarification, please indicate the arguments that are still unclear.
>
>
> Section 3 and 4 (major concerns)
>
> Unlike section 2 which was pretty step by step, sections 3 and 4 read more like a list of known facts patched together. It is very hard to figure out what was the goal and how the neural network comes into play. I think given that this is an ML journal and not a PDE journal, a clearer roadmap is needed, and to actually show what role that , both in training and modeling, is giving.
>
>     Thank you for the remark. I would appreciate some additional clarification on what to improve. For now, I have made the following modifications:
>         - I added a paragraph after the first paragraph of section 3 to motivate the use of the sublinear networks a bit further.
>         - I have brought back some elements of the appendix into the main text, concerning the computation of geometric-differential quantities. I hope it makes this part more self-contained and clear.
>
>
> I was not clear as to how is being trained. Where is the objective function? Is it constrained? I saw that L-BFGS is used, but to optimize what exactly?
>
>     Thank you for your remark. I made all the loss functions more visible, and used the notation $L$ for each one of them. I also specified more explicitly the loss functions in the case of the Poisson problems and the computation of the Mahler volume. Concerning the constraint question, I added the sentence "Contrary to classical numerical convex shape optimization, every shape optimization problems considered here reduces to an unconstrained optimization problem".
>
>
> If numerics is an important contribution, there should be more baselines to compare against.
>
>     As is stated in the first paragraph of Section 4: "A direct quantitative comparison with classical methods of convex shape optimization is not possible in practice, as no publicly available implementations exist in a form that would allow a consistent and reproducible evaluation". Most of the code for tackling this kind of problem is not open source, and may be incomplete when it is. I would be glad to compare to existing baseline if I missed some runnable, open-source code.
>
> Overall
>
> It's confusing to use for two separate functions. Suggest splitting them into two symbols.
>
>     It is fixed, hopefully it makes the proofs and overall paper clearer.

---

> > ### Comment · Reviewer_Hq8A · 2026-07-14
> > **Initial thoughts**
> >
> > I plan on giving the revision a much more detailed read in the next couple days and will post more comments then, but I took a quick skim and have some comments in the meantime:
> >
> >  - You still need maybe 2-4 sentences, very easy to find in section 1 or beginning of section 3 (or both), that 1. clearly defines what is shape optimization and 2. why it is relevant to the machine learning community. For example, you can tie it to object tracking or point clouds from sensors, and then add a few application survey papers as citations. Without a strong application grounding, I don't think readers will buy that this is a machine learning paper and not just an applied math paper.
> >
> >  - In the beginning of section 2, you should introduce clearly the objective function to be minimized. You don't have to give precise notation, and can even describe in words, but before we are subjected to a single mathematical fact we need a clear roadmap. In the beginning of section 3, there should be a clear mathematical formulation of the objective function.
> >
> >  - I noticed you actually did this with two citations, Schneider (2013), Huang 2025 for Minkowski problem, and Evans and Gariepy 2025 for change for variables. There is no way that fundamental fields like these were developed in 2013 or 2025; my effort in giving you some citations that were pre-2000 was to demonstrate this. The standard is to give credit, to the best of your ability, to the work that presented the fact as a novelty. If it is not possible to do this (I guess change of variables would have to be attributed to Gauss and honestly does not need to be cited) then you can casually say "this is a well-known mathematical fact" and not give a citation (or not mention citing at all), but it reads very strange and misleading to cite someone's recent textbook, of which there are many on the market for every topic.
> >
> > Also, my reason to give many citations for gauges is that gauges is the main focus of your study. I think for that one, you need many more citations than I provided even, to demonstrate a serious study.

---

> > > ### Comment · Reviewer_Hq8A · 2026-07-16
> > > **detailed review**
> > >
> > > Ok, most of my issues about section 2 have been resolved. A few more that I think can be easily handled:
> > >
> > >  - For thm 2.4, and then used in the proof of Thm 2.2, I think there's a typo. the $m$ in the exponent should be $d$ maybe?
> > >  - In the proof for Thm 2.2, $\delta$ is produced but never explained. Is this the result of a concentration inequality?
> > >
> > > Section 2 is where I am most familiar with the technical details, so at this point, in terms of technical stuff, I am satisfied.
> > >
> > > I also reread the introduction and other sections and feel there is more of an attempt to connect this work with the broader audience, e.g. clearer problem statements and motivations.
> > >
> > > I do agree with other reviewers on the subject of related works and baselines however. In both cases there seems to be extensive work on shape optimization. It seems that this is somewhat discussed in various parts of the paper, but 1. no clearly consolidated "related works" section exists in the introduction and 2. no clear comparison between this method and those methods exist in a consolidated way. Maybe it requires too much setup to do an apples to apples comparison on the same problem bank, but a table that somewhat qualitatively compares things is still useful, e.g. "Method A, requires $m = O(1/\epsilon^2)$ samples and polynomial in runtime" or something like that, comparing the different methods against yours, and against the PINNs comparison also shown in the appendix. (By the way, maybe it should also be showcased in the main paper, to present some sort of comparison.)